# Superseding Model Scaling by Penalizing Dead Units and Points with Separation Constraints

## Abstract

In this article, we study a proposal that enables to train extremely thin (4 or 8 neurons per layer) and relatively deep (more than 100 layers) feedforward networks without resorting to any architectural modification such as Residual or Dense connections, data normalization or model scaling. We accomplish that by alleviating two problems. One of them are neurons whose output is zero for all the dataset, which renders them useless. This problem is also known as *dead neurons*. The other is a less studied problem, *dead points*. Dead points refers to data points that are mapped to zero during the forward pass of the network. As such, the gradient generated by those points is not propagated back past the layer where they die, thus having no effect in the training process. In this work, we characterize both problems and propose a constraint formulation that added to the standard loss function solves them both. As an additional benefit, the proposed method allows to initialize the network weights with constant values or even zero and still allowing the network to converge to reasonable results. We show very promising results on a toy, MNIST, and CIFAR-10 datasets.

## 1 Introduction

The success of Deep Neural Networks (DNN for short) is linked to its ability to learn *abstract representations* from input data in a hierarchical fashion (LeCun et al. (2006); Ramachandran et al. (2017)). However, the concepts of depth and width in networks are often used as instrumental elements to address different DNN pathologies during the learning process. Examples of these are: *vanishing gradient* (Hochreiter (1991; 2001)), *exploding gradient* (Pascanu et al. (2013)), *dead units* (Maas (2013); Douglas & Yu (2018); Guerraoui et al. (2017)), or the *degradation problem* (He et al. (2015b)).

In order to address the former issues, we find many methods and techniques that we can roughly classify in two families: *data manipulation* and *architectural modifications*. The most commonly used data manipulation technique is data normalization on the output of the layers, for example using batch normalization (Ioffe & Szegedy (2015)). Examples of architectural modifications include the use of additional connections, as done in ResNets (He et al. (2015b)) or DenseNets (Huang et al. (2016)); unit augmentation, as in *leaky*-`ReLU` (Maas (2013)), `PReLU` (He et al. (2015a)), `C-ReLU` (Shang et al. (2016)), or linked neurons (Riera & Pujol (2017)); or the increment of layer's width with depth (Zagoruyko & Komodakis (2016); Szegedy et al. (2014)). This last widely used approach effectively increases the the size of the network requiring larger computational power. This has led to several works (Hasanpour et al. (2018); Tan & Le (2019)) that address it by offering heuristics on how to scale the network. Recent research (Liu et al. (2018)) suggests that this inverted pyramidal architecture is not optimal.

Besides of the width scaling strategy, the concept of dead units[1] is of particular interest to this work. A dead neuron is defined as the neuron with a constant or zero output for all training data points. This effectively renders this unit ineffective during the learning process. In (Lu et al., 2019) it is shown that as the depth increases the probability of finding dead neurons also increases, to the point

---

[1]In this work we use *unit* and *neuron* indistinctly.

that the entire network can be dead even at initialization. Additionally, as expected, as the width of a layer increases, the probability of having dead neurons decreases.

In this article we characterize another pathology, the dead point. The dead point is a dual concept to the dead neuron. A dead point corresponds to a data point that does not reach the output of the network. This is, the activation is zero for all the units in a given layer. As a result, this data point will have no influence in the training process. As in the case of dead units, both condition are not recoverable by back-propagation. This virtually reduces the size of the training set.

In order to solve the former issues, we propose and present a geometrical optimization constraint that is added to the loss function. As a result of adding this constraint we ensure that all neurons are active and no points are dead. The rationale behind the geometrical constraint is to control that all units/neurons and data points are alive by constraining pre-activation values in such a way that the hyperplanes associated to the non-linearity of each ReLU neuron separates, at least, one data point from the rest of the dataset. This has the additional effect of enabling the learning process to propagate all information through all the network avoiding the instrumental need of using additional connections or inverted pyramidal architectures.

We test our proposal in a series of controlled experiments to showcase the effect of applying the proposed *Separation Constraints*. We find that we can arbitrarily increase the network depth using the same constant width when compared to standard feedforward network and Batch Normalization Ioffe & Szegedy (2015) networks even when the width of the layer is extremely small (4 or 8 neurons). We additionally provide evidences that using this same approach we can initialize network parameters to zero and still achieve reasonable performance. These promising results gives insight on learning dynamics and suggest potential lines for future checks and research.

The article is organized as follows: Section 2 introduce the characterization of dead neurons and dead points, Section 3 introduces the two geometrical constraints that ensure that neurons and data points are alive, Section 4 describes the experiments and result, and, finally, Section 5 concludes the paper and suggests future lines of research.

## 2    CHARACTERIZING DEAD NEURONS AND DEAD POINTS

A standard feed-forward ReLU DNN (LeCun et al., 2015) $F$ can be formally written as a multi-valued real function, $F(\mathbf{x})$, that is created by composing a collection of $D$ vector *layer* functions $\ell : \mathbb{R}^{n_{k-1}} \to \mathbb{R}^{n_k}$. Layer $k$ is defined as the sum of a collection of scalar functions (*units*):

$$\ell_k(\mathbf{x}) = \sum_{j=1}^{n_k} u_j^k(\mathbf{x})\hat{\mathbf{e}}_j \tag{1}$$

that affinely depend on a weight vector $\mathbf{w}_j^k \in \mathbb{R}^{n_k}$ and a bias parameter $b_j^k \in \mathbb{R}$. When using rectified linear units this value is truncated on negative values:

$$u_j^k(\mathbf{x}) = \max(0, \mathbf{w}_j^k \cdot \mathbf{x} + b_j^k). \tag{2}$$

Considering the hyperplane defined by $\mathbf{w}_j^k \cdot \mathbf{x} + b_j^k = 0$, each unit defines a partition of the space $\mathbb{R}^{n_k}$ in two sets: the *upper* part of unit $u_j^k$ and the *lower* part of $u_j^k$:

$$
\begin{aligned}
upper(u_j^k) &= \{\mathbf{x} : \mathbf{w}_j^k \cdot \mathbf{x} + b_j^k > 0\} \\
lower(u_j^k) &= \{\mathbf{x} : \mathbf{w}_j^k \cdot \mathbf{x} + b_j^k \leq 0\}
\end{aligned} \tag{3}
$$

We define the *affine* component of a layer function $\ell_k$ as the intersection of the upper parts of its units, and its zero set as the intersection of the lower parts, as follows:

$$A(\ell_k) = \bigcap_{j=1}^{n_k} upper(u_j^k), \quad Z(\ell_k) = \bigcap_{j=1}^{n_k} lower(u_j^k) \tag{4}$$

**Remark 2.1** (Dead unit). *In a given ReLU-DNN $F : \mathbb{R}^n \to \mathbb{R}^k$, we say that the j-th unit of layer $\ell_k$, $u_j^k$ is* dead *with respect to a data set $\mathbb{X} \subset \mathbb{R}^{n_k}$ if and only if*

$$\mathbb{X} \subset lower(u_j^k). \tag{5}$$

Observe that if a unit is dead, the output of the unit will be zero for the entire dataset, rendering the unit useless. Moreover, since the gradient is zero as well, it remains in this state for the rest of the training (See Lu et al. (2019); Shin & Karniadakis (2019)). This effectively reduce the network learning capacity.

**Remark 2.2** (Dead point with respect to a layer). *Given* ReLU*-DNN* $F : \mathbb{R}^n \to \mathbb{R}^k$*, we say that a point* $\mathbf{x} \in \mathbb{X}$ *is* dead *with regards to layer* $\ell_k$ *if*

$$\mathbf{x} \in Z(\ell_k). \tag{6}$$

**Remark 2.3** (Dead point). *In particular, if* $\mathbb{X} \subset \mathbb{R}^n$*, we say that point* $\mathbf{x} \in \mathbb{X}$ *is* dead *with respect to a network* $F$ *with depth* $D$ *if it gets mapped to the zero set of a layer in its transit through the network. This is*

$$(\exists k, 1 < k \leq D | \ell_{k-1} \circ \ldots \ell_1(\mathbf{x}) \in Z(\ell_k)). \tag{7}$$

Any dead point fulfilling Equation 7 will show zero gradient in the layers previous to $\ell_k$. This hinders the learning process by effectively reducing the data set size. Again, this condition is not reversible using standard back-propagation. In a similar fashion to the case with dead units, the probability of finding a dead point increases with network's depth and decreases with layer's width.

## 3 Introducing separability constraints

In this section we introduce the desiderata for units and points to remain alive. Then, we proceed to formulate the separability contraints that fulfill the desired conditions.

Let us introduce the concept of a *separating* unit with respect to an arbitrary set $\mathbb{X}$.

**Definition 3.1** (Separating Unit). *Given an arbitrary set* $\mathbb{X} \subset \mathbb{R}^{n_k}$*, we say that the j-th unit on layer* $k$*,* $u_j^k$*, is able to* separate *through* $\mathbb{X}$ *if the following predicate is satisfied:*

$$R_{\mathbb{X}}(u_j^k) \equiv \emptyset \neq \{lower(u_j^k) \cap \mathbb{X}\} \subset \mathbb{X} \tag{8}$$

Thus, by construction, a separating unit can not be dead, and if $R_{\mathbb{X}}(u_j^k)$ is valid, $u_j^k$ can not degrade set $\mathbb{X}$ to zero. In other words, this condition ensures that each unit always separates at least one data point.

In terms of points, we can define a *separating point* as follows:

**Definition 3.2** (Separating point). *Given an arbitrary set* $\mathbb{X} \subset \mathbb{R}^{n_k}$*, we say that point* $\mathbf{x} \in \mathbb{X}$ *is* separating *a layer function* $\ell_k$ *if there exist indices* $j, l \in \{1, \ldots, m_k\}$ *for which the following predicate is satisfied.*

$$R_k^x(j, l) \equiv x \in \{upper(u_j^k) \cap lower(u_l^k)\} \tag{9}$$

Again, by construction, a separating point ensures that each point in the data set has at least one unit in each layer with an activation different to zero and another with an activation equal to zero.

### 3.1 Modelling Unit-based Separation Constraint (Sep-U)

Unit based separation contraint (Sep-U) is designed to model predicate 8 with the goal of avoiding the presence of dead units.

Given a unit $u_j^k$ in layer $\ell_k : \mathbb{R}^{n_k} \to \mathbb{R}^{n_{k+1}}$ from a ReLU DNN $F$ of depth $D$, predicate 8 can be simply modelled imposing the following constraints,

$$\begin{aligned} \max_{i=1,\ldots,|\mathbb{X}|} \{\mathbf{w}_j^k \cdot \mathbf{x}_i + b_j^k\} > 0 \\ \min_{i=1,\ldots,|\mathbb{X}|} \{\mathbf{w}_j^k \cdot \mathbf{x}_i + b_j^k\} < 0 \end{aligned} \tag{10}$$

These strict inequalities can not be directly optimized. It is easily to see that these can be rewritten as

$$\max_{i=1,\ldots,|\mathbb{X}|}\{\mathbf{w}_j^k \cdot \mathbf{x}_i + b_j^k\} \geq 1$$
$$\min_{i=1,\ldots,|\mathbb{X}|}\{\mathbf{w}_j^k \cdot \mathbf{x}_i + b_j^k\} \leq -1 \tag{11}$$

The use of the former constraints makes most problems unfeasible. Thus, in a similar fashion to soft-margin SVM (Cortes & Vapnik, 1995), we introduce a set of positive slack variables $\{\xi_{jk}^+, \xi_{jk}^-\}$ that account for constraint violations as follows,

$$\max_{i=1,\ldots,|\mathbb{X}|}\{\mathbf{w}_j^k \cdot \mathbf{x}_i + b_j^k\} \geq 1 - \xi_{jk}^+,$$
$$\min_{i=1,\ldots,|\mathbb{X}|}\{\mathbf{w}_j^k \cdot \mathbf{x}_i + b_j^k\} \leq -1 + \xi_{jk}^-, \tag{12}$$
$$\xi_{jk}^+, \xi_{jk}^- \geq 0,$$

for $k = 1, \ldots, D$ and $n_k$ the number of units of layer $k$. The intuition behind the introduction of these constraints is as follows: by minimizing $\xi^+$ at least one pre-activation value is forced to be greater (or equal) than 1. Simmetrically, minimizing $\xi^-$ promotes at least one pre-activation to be below $-1$. This effectively fulfills Predicate 8 and penalize the apparition of dead units. We show how to differentiate the Separation Constraints with regards to the parameters in the Appendix Section A.3.

At a global network scale, we can aggregate all the slack variable in a single optimization objective as follows,

$$g_U(\boldsymbol{\xi}^+, \boldsymbol{\xi}^-) = \frac{1}{2}\sum_{k=1}^{D}\sum_{j=1}^{n_k}(\xi_{jk}^+ + \xi_{jk}^-). \tag{13}$$

### 3.2 Modelling Point Based Separation Constraint (Sep-P)

The derivation of the Point Based Separation Constraints (Sep-P) follows a parallel process to Sep-U. In order to avoid the presence of dead points it suffices to fulfill Predicate 9. Similarly to the former derivation, we introduce a set of slack variables for each each point on the batch. That is, given $\mathbf{x}_i \in \mathbb{X}$, and $u_1^k, \ldots, u_n^k$ unit functions in a layer $\ell_k$, we define slack variables $\xi_{ik}^-, \xi_{ik}^+$ in the context of the following constraints,

$$\max_{j=1,\ldots,n_k}\{\mathbf{w}_j^k \cdot \mathbf{x}_i + b_j^k\} \geq 1 - \xi_{ik}^+,$$
$$\min_{j=1,\ldots,n_k}\{\mathbf{w}_j^k \cdot \mathbf{x}_i + b_j^k\} \leq -1 + \xi_{ik}^-, \tag{14}$$
$$\xi_{jk}^+, \xi_{jk}^- \geq 0.$$

Observe that the minimization of the slacks makes that for any data point at least one activation is above 1 and another is below -1.

We can summarize all the point-based slack variables in a single optimization objective as follows,

$$g_P(\boldsymbol{\xi}^+, \boldsymbol{\xi}^-) = \frac{1}{2}\sum_{k=1}^{D}\sum_{i=1}^{|\mathbb{X}|}(\xi_{ik}^+ + \xi_{ik}^-). \tag{15}$$

### 3.3 Training with Separating Constraints

The new optimization objectives can now be added to the original loss objective using a simple scalarization (Boyd & Vandenberghe (2004)) as follows,

$$\operatorname*{arg\,min}_{\theta,\xi^+,\xi^-} \mathcal{L}(\mathcal{T},\theta) + \lambda\big(g_U(\boldsymbol{\xi}^+, \boldsymbol{\xi}^-) + g_P(\boldsymbol{\xi}^+, \boldsymbol{\xi}^-)\big), \tag{16}$$

where $\lambda$ is a hyper-parameter that introduces a trade-off between the constraint fulfillment and the main loss function.

In terms of memory *complexity*, the former constraints introduce a very small memory overhead. In particular, `Sep-U` places a pair of constraints on each of the units of the network, so the complexity with respect to `Sep-U` scales with the size of the network as $2\sum_{k=1}^{D} n_k$. Alternatively, `Sep-P` places a pair of constraints in each of the points of the data set or selected subset $\mathbb{X}$. In practice, one can use $\mathbb{X}$ as the training batch. Thus, the memory complexity scales with the size of the batch and the number of layers, i.e. $2D|\mathbb{X}|$. Furthermore, since the resulting gradient of both types of constraints depends only on the input of the layer that is already computed in the forward pass, we only add the cost of storing the slacks. Therefore, the total cost in terms of number of constraint is the addition of the former terms, i.e. $2\sum_{k=1}^{D} n_k + 2D|\mathbb{X}|$.

## 4 EXPERIMENTS AND RESULTS

In this section we explore the application of the proposed constraints in different datasets. For that task we train all methods with different choices of depth and width parameters. The network architecture used is *rectangular*, i.e. networks with a fixed layer width for all the layers.

**Datasets:** Due to the large amount of computational resources required for the depth and width grid training, we are forced to chose three controlled datasets in our experimentation: the `MOONS` dataset (sampling 100 points, 85 for training and 15 for validation), the `MNIST` dataset as described in LeCun and Cortes. LeCun & Cortes (2010) and the `CIFAR-10` dataset described in Krizhevsky (2009).

**Experimental setting:** We compare the combination of `Sep-U` and `Sep-P` (`Sep-UP` from now on) to feed-forward `ReLU` networks (Glorot et al., 2011) and Batch Normalization as described in Ioffe & Szegedy (2015) using the same architecture.

For the `MOONS` dataset, we use depths from 1 to 120 in steps of 10, and width from 1 to 25 in steps of 1 between 1 and 5, and steps of 5 afterwards. In the case of the `MNIST` dataset, we use depth from 2 to 64 and width from 2 to 8 in steps of 4. Finally, for `CIFAR-10` we use depths in $\{2, 10, 25, 30, 40\}$ with a fixed width of 10 due memory constraints.

**Training Parameters:** All the networks used were optimized using Adam (Kingma & Ba, 2014). More specifically, for the `MOONS` dataset we used a learning rate of $0.01$ for 5000 epochs and a batch size of 85. Meanwhile, for both `MNIST` and `CIFAR-10` we used a learning rate of $0.0001$ for 50 epochs and a batch size of 1024. We used convolutional layers with a $(3, 3)$ kernel size for both `MNIST` and `CIFAR-10`. We used $\lambda_{MOONS} = 10^{-4}$ $\lambda_{MNIST} = \lambda_{CIFAR} = 10^{-8}$ for `Sep-UP`. Our experiments were conducted using `Keras` (Chollet et al., 2015) and `TensorFlow` (Abadi et al., 2015), fixing the random seed to an arbitrary value of 10.

As initialization scheme, we used Glorot uniform from Glorot & Bengio (2010) for all the methods and datasets.

### 4.1 RESULTS

Figure 1 shows the results obtained in the `MOONS` dataset. Our proposal `Sep-UP` is able to train networks successfully without increasing the width up to 60 layers deep (see Figures 1c and 1f), while `ReLU` breaks down at only 30 layers (see Figures 1a and 1d) and `ReLU + BN` suffers from severe accuracy degradation (see Figures 1b and 1b).

Observe that no configuration with lower width than 2-3 is successful in achieving maximum accuracy. We understand that there exists a minimum width required and this is related to the complexity of the problem. When using wider layers, the rest of the width is instrumentally used to enable the training of deeper networks. As previously commented, the larger the layer's width, the higher the chances of finding active units that do not cause dead points and dead units during initialization. In opposition, `Sep-UP` succesfully overcomes that constraint.

Notice that though `Sep-UP` is superior to all its competitors, it starts showing performance degradation after reaching depth 60 with the minimum width of 3. Considering that the number of parameters increases with the depth and width of the network and that we have a finite number of trained epochs, we conjecture that the displayed degradation is strictly due to the lack of convergence of the constrained network.

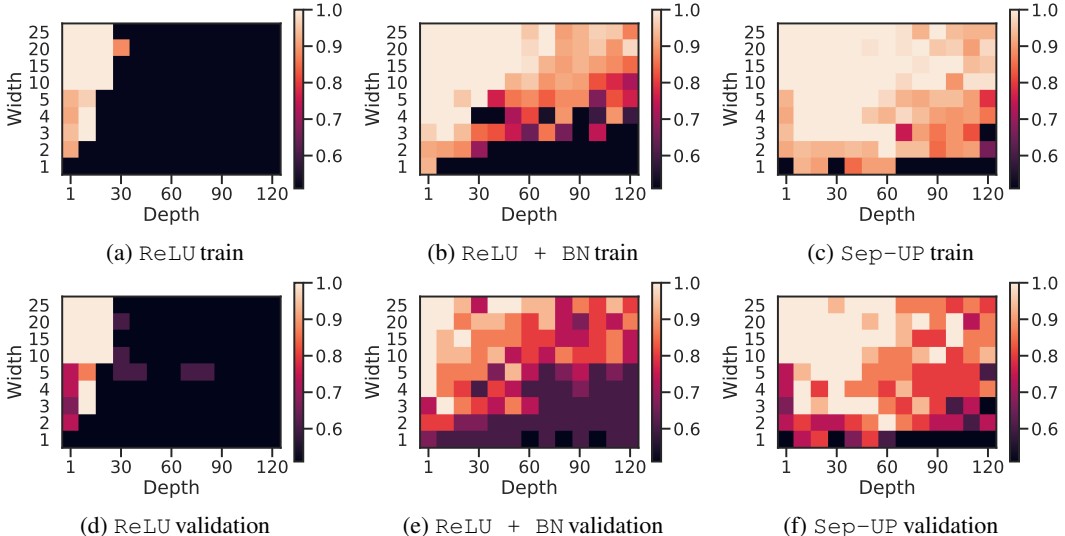

Figure 1: Depth vs width accuracy heatmap a for a grid of rectangular networks with width from 1 to 25 and depth from 1 to 120, trained using Adam with a learning rate of 0.01 in the MOONS dataset for 5000 epochs. The color shows the accuracy attained of each of the combinations of width and depth, with clear beige at 1 and black at 0.5. Notice how ReLU breaks down at 20 layers and ReLU + BN requires more units per layer as increasing depth, while Sep-UP works with the minimum width (3) up to 60 layers deep.

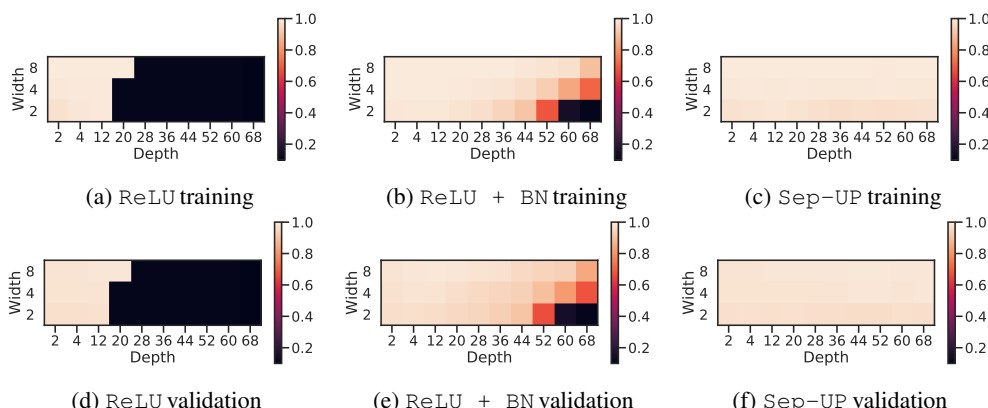

Figure 2: Depth vs width accuracy heatmap a for a grid of rectangular networks with width from 2 to 8 and depth from 2 to 68, trained using Adam with a learning rate of 0.0001 in the MNIST dataset for 50 epochs. The color shows the accuracy attained of each of the combinations of width and depth, with clear beige at 1 and black at 0.1. Notice how ReLU breaks down at 20 layers and ReLU + BN accuracy degrades with depth, while Sep-UP shows constant accuracy disregarding the number of layers.

The separation constraint also proves successful on convolutional networks, as tested in MNIST and CIFAR-10 datasets. Figure 2 shows a similar behaviour to the MOONS dataset (Figure1). ReLU breaks down after a few layers, ReLU + BN delays the degradation of accuracy, while Sep-UP remains functional regardless of the depth. In the case of our experiments with the CIFAR-10 dataset (as presented on Figure 3) all the methods degrade with depth, but Sep-UP is the most robust. In regards to accuracy ReLU performs best closely followed by Sep-UP, while ReLU + BN clearly overfits. The poorer accuracy values shown are due to a limited choice of width, clearly inferior to the minimum required by the dataset, and a potential lack of convergence of the separating constraints.

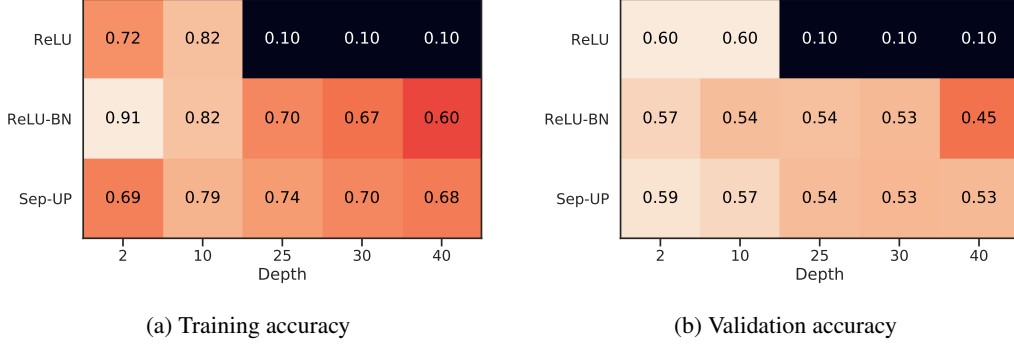

(a) Training accuracy                    (b) Validation accuracy

Figure 3: Depth vs Width accuracy heatmap a for a grid of rectangular networks with width 10 and depth from 2 to 40, trained using `Adam` with a learning rate of 0.0001 in the `CIFAR-10` dataset for 50 epochs. The color shows the accuracy attained of each of the combinations of width and depth, with clear beige at 1 and black at 0.1. Observe how `Sep-UP` shows inferior degradation in accuracy as depth increases compared to `ReLU + BN`.

Additional results in the Appendix further elaborate the contributions of each term `Sep-U` and `Sep-P`, independently.

### 4.2 RESULTS USING ZERO INITIALIZATION

In order to test the invariace to initialization scheme of `Sep-UP`, we use Zero Initialization. As its name states, in this initialization scheme all weights and biases are set to zero. However, a small variation of the scheme must be introduced in order to break symmetry for the constraints to apply. Since all the units are initialized to the same value (zero), we use Annealed Dropout (Rennie et al., 2014). Additionally, instead of adding $\xi^+$ and $\xi^-$ pairs as in Equations 13 and 15, we use a convex combination with a small perturbation $\rho$. In our experiments, we use a value of $\rho = 0.01$.

$$(\frac{1}{2} + \rho)\xi^+ + (\frac{1}{2} - \rho)\xi^- \tag{17}$$

Figure 4 summarizes the results found using zero initalization with our constraint formulation. In comparison to Glorot, we observe that zero initalization requires wider networks. Indeed, at a depth of 60, Zero initialization requires a width of 25 units while the Glorot scheme work with only 2 units. However, if we contrast `ReLU + BN` to our constraint formulation we find that it is unable to train networks of past depth 50, while Zero initialization achieves 70 layers.

Figure 4 shows the results using zero initialization. Although reported values and behavior are slightly worse compared to Glorot initialization, our results are promising. Notice how at depth of 60 layers, the zero initialization requires a width of 25 units in contrast to 2 required for Glorot. In addition, zero initialization rquires only 70 layers to reach perfect accuracy in contrast to the 100 required in Glorot (see Figures 1c and 1f). Moreover, in comparison to `ReLU + BN`, zero initialized networks show superior performance and behavior (check Figures 1b and 1e).

## 5 CONCLUSIONS

Through the Separation Constraints, we have shown that deeper networks can be trained without increasing the width of the layers. Moreover, this increment can be done using very small width values. In this sense we consider that effective training of deeper networks can be achieved by better accommodating the network to the input data. This departs from many proposal that achieve similar effects by modifying the architecture of the network or manipulating the data. We believe that this work shows an alternative research path in the pursuit of effective and efficient learning techniques for deep neural networks. This also opens the possibility of avoiding the use of pyramidal

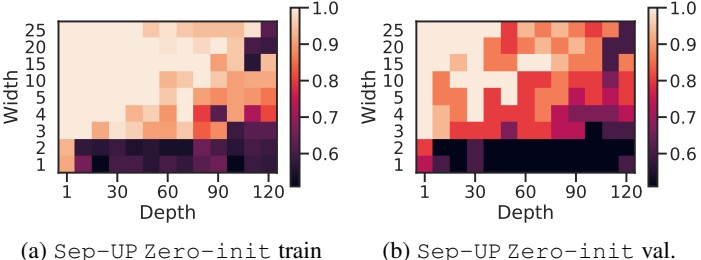

(a) `Sep-UP Zero-init` train        (b) `Sep-UP Zero-init` val.

Figure 4: Depth vs width accuracy heatmap a for a grid of rectangular networks with (width from 1 to 25 and depth from 1 to 120), trained using `Adam` using a learning rate of 0.01, over the `MOONS` dataset (5000 epochs and Zero Initialization). The color shows the accuracy attained of each of the combinations of width and depth, with clear beige at 1 and black at 0.5. Notice how although being inferior to Glorot (Figures 1c and 1f) is superior to `ReLU + BN` (Figures 1b and 1e).

architectures commonly employed in DNNs, thus removing the computational burden caused by the additional units and reducing the dimensionality of the internal representations.

We additionally show that our proposal enables the use of Zero Initialization. The results are promising, and though further experimentation is still needed, they still manage to surpass Batch Normalization in terms of depth, width, and accuracy. Nevertheless, the modifications used to break the symmetry of constraints and units need further consideration in order to achieve the same performance than random initialization.

Despite this article provides no information about the dynamics of the training process using Separation Constraints, preliminary revision shows interesting properties. In particular, training displays slight instabilities which appear as the units/points are reactivated. Although this does not hinder the training process, further study is needed in order to guarantee a smoother convergence.

The extension of this work to other activation functions is still to be explored. However, we conjecture that in cases where the activation function display a flat region, e.g. ELU, tanh, or even sigmoid, the current proposal can be applied with minor changes.

Finally, while the separation constraints prevent the vanishing gradient effect, the *exploding gradient* problem still remains. Extending the Separation Constraint with an upper bound on the magnitudes of the pre-activation, similar to $\epsilon$-insensitive loss, might address it. It could be also helpful to explore other activation functions whose gradient vanishes with high pre-activations, such as the logistic family.

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

# A    APPENDIX

## A.1    UNIT BASED AND POINT BASED SEPARATION CONSTRAINT EXPERIMENT

In order to justify the decision of combining `Sep-U` and `Sep-P` into `Sep-UP`, we provide Figure 5 showing the same experiment than Figure 1 for `Sep-U` and `Sep-U`. We find how although `Sep-U` is able to train deeper and thinner networks it suffers from inferior accuracy (see Figures 5a and 5c), requiring the use of the layer width increase technique. Contrarily, `Sep-P` is able to train without increasing the width from 1 to 40 layers deep, but it breaks down afterwards. Provided how complementary are their strengths and weaknesses, it is sensible to combine them, resulting in the superior performance (larger constant width area together with greater accuracy in higher depths) of `Sep-UP`, see Figures 1c and 1f.

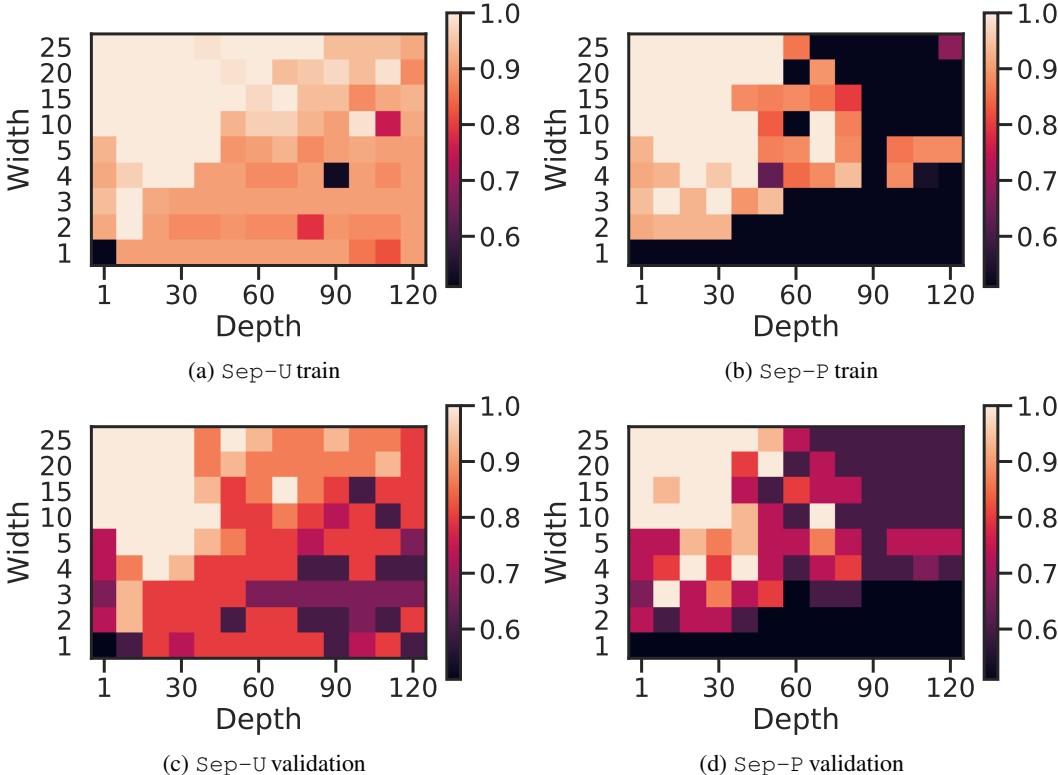

(a) `Sep-U` train

(b) `Sep-P` train

(c) `Sep-U` validation

(d) `Sep-P` validation

Figure 5: Depth vs width accuracy heatmap a for a grid of rectangular networks with width from 1 to 25 and depth from 1 to 120, trained using `Adam` with a learning rate of 0.01 in the `MOONS` dataset for 5000 epochs. The color shows the accuracy attained of each of the combinations of width and depth, with clear beige at 1 and black at 0.5. Notice how `Sep-U` enables train thinner and deeper networks (120 layers of single unit), but with reduced performance compared to `Sep-UP`. It also requires increasing the width of the layers in order to achieve perfect accuracy as `ReLU + BN`. Alternatively, `Sep-P` shows a constant width area between 1 and 40 layers, but its performance is much worse above. This justifies the decision of combining them both into `Sep-UP` to combine their strengths.

## A.2    PROBABILITY OF A POINT TO BE DEAD IN A NEURAL NETWORK

We conjecture that difficulty arising when training neural network as the depth increases or the width decreases, is due the incorrect positioning of the hyperplanes of the units after initialization which causes the apparition of dead points (recall Remark 2.3), that ultimately break back-propagation. In this section we offer a formal treatment of this idea.

**Theorem A.1** (The probability of a point to be dead increases with depth and decreases with width). *Let $F : \mathbb{R}^{n_1} \to \mathbb{R}^{n_D}$ be a feed-forward neural network composed by a collection of $D \in \mathbb{N}$ layer functions $\ell : \mathbb{R}^{n_{k-1}} \to \mathbb{R}^{n_k}$ with $k \in 1, \ldots, D$, we denote $D$ as the* depth *of the network. The layers are composed by a collection of $n_k \in \mathbb{N}$ ReLU unit functions $u : \mathbb{R}^{n_{k-1}} \to \mathbb{R}_+$, we denote $n_k$ as the* width *of the layer. Let the network be initialized following a random distribution. Then, given a point $\mathbf{x} \in \mathbb{R}^{n_1}$ we say that the probability of the point dying (Recall Definition 7) increases with $D$ and decreases with $n_k$.*

**Lemma A.2** (Probability of a point to be dead for a ReLU unit follows Bernoulli). *Let be a point $\mathbf{x} \in \mathbb{R}^{n_1}$ and a ReLU unit $u$ parametrized by a weight vector $\mathbf{w} \in \mathbb{R}^n$ and a bias $b \in \mathbb{R}$ initialized following a random distribution, then the probability of point $\mathbf{x}$ being dead for $u$ follows a* Bernoulli *distribution of unknown parameter $p$.*

$$P(\mathbf{x} \in lower(u)) \sim Bernoulli(p) \tag{18}$$

*Proof.* Since the unit is initialized randomly according to a given distribution (i.e. normal, uniform, Glorot, ...), the probability of its hyper-plane parametrized by $\mathbf{w}, b$ facing $\mathbf{x}$ is an unknown $p$ according to the distribution used. Therefore, the probability of $x$ belonging to $lower(u)$ (Recall Equation 3) must be $p$.

As the lower set (Recall Definition 3) pre-activation is truncated, its gradient with respect the parameters will be zero (See Rey-Torres et al. (2019, Section 2)). Therefore, any point belonging to $lower(u)$ will be dead in regards to $u$. $\square$

**Lemma A.3** (Probability of a point to be dead for a ReLU layer). *Let $\ell_k$ be a layer composed of $n_k \in \mathbb{N}$ ReLU units $u$ initialized following the same random distribution. Let be a point $\mathbf{x} \in \mathbb{R}^{n_{k-1}}$. Then the probability of $\mathbf{x}$ being dead for $\ell_k$ is:*

$$P(\mathbf{x} \in Z(\ell_k)) = P(\mathbf{x} \in lower(u))^{n_k} \tag{19}$$

*Proof.* Similarly to the unit case, points belonging to the zero set of a layer (Recall Definition 4) have zero gradient with respect the parameters of the layer due the truncation (See Rey-Torres et al. (2019, Section 2)).

The probability of a point belonging to the zero set of a layer $P(\mathbf{x} \in Z(\ell_k))$ is equal to the probability of belonging to all the lowers of the units of the layer $P(\forall u \in \ell | \mathbf{x} \in lower(u))$. Since unit parameters are sampled independently during initialization and the probability of a point being dead for a unit follows Bernoulli, then the probability of a point belonging to $Z(\ell_k)$ is the product of the probabilities of belonging to the lower of each of the units. $\square$

**Lemma A.4** (The probability of a point to be dead for a layer diminishes with its width). *Let $\ell_k$ be a layer composed of $n_k \in \mathbb{N}$ ReLU units $u$ initialized following a random distribution. Let be a point $\mathbf{x} \in \mathbb{R}^{n_{k-1}}$. Then the probability of $\mathbf{x}$ being dead diminishes with $n_k$:*

$$\lim_{n_k \to \infty} P(\mathbf{x} \in Z(\ell)) = 0 \tag{20}$$

*Proof.* The probability of a point being dead for a layer is the multiplication of the probabilities of being dead of each of its units, recall Lemma A.3. Therefore, as the probabilities are lower than 1 since the upper set of a unit is never empty, in the limit where $n_k$ tends to infinity the probability tends to zero. $\square$

Note how Lemma A.4 can explain why increasing width allows us to train deeper networks (See Hasanpour et al. (2018); Huang et al. (2016)), it simply improves the chances of finding a hyper-plane that allows the points to live.

**Definition A.1** (Dead point set in a neural network). *Let $F : \mathbb{R}^{n_1} \to \mathbb{R}^{n_D}$ be a feed-forward neural network composed by a collection of $D \in \mathbb{N}$ layer functions $\ell : \mathbb{R}^{n_{k-1}} \to \mathbb{R}^{n_k}$ with $k \in 1, \ldots, D$, we denote $D$ as the* depth *of the network. The layers are composed by a collection of $n_k \in \mathbb{N}$ ReLU unit functions $u : \mathbb{R}^{n_{k-1}} \to \mathbb{R}_+$, we denote $n_k$ as the* width *of the layer. Then, we define the set of dead points of a neural network following Equation 7 as.*

$$Z(F) = \{\mathbf{x} | (\exists k | 1 < k \le D : \ell_{k-1} \circ \ldots \ell_1(\mathbf{x}) \in Z(\ell_k)) : \mathbf{x}\} \tag{21}$$

**Lemma A.5** (Probability of a point to be dead for a `ReLU` feed-forward network). *Let $F : \mathbb{R}^{n_1} \to \mathbb{R}^{n_D}$ be a feed-forward neural network composed by a collection of $D \in \mathbb{N}$ layer functions $\ell : \mathbb{R}^{n_{k-1}} \to \mathbb{R}^{n_k}$ with $k \in 1, \ldots, D$, we denote $D$ as the* depth *of the network. The layers are composed by a collection of $n_k \in \mathbb{N}$ `ReLU` unit functions $u : \mathbb{R}^{n_{k-1}} \to \mathbb{R}_+$, we denote $n_k$ as the* width *of the layer. Let the network be initialized following a random distribution. Then, given a point $\mathbf{x} \in \mathbb{R}^{n_1}$ we say that the probability of the point dying (Recall Definition 7) is*

$$P(\mathbf{x} \in Z(F)) = 1 - \prod_{i=1}^{D}(1 - P(\mathbf{x} \in Z(\ell_i))) \tag{22}$$

*Proof.* The probability of not dying in a given layer is equal to 1 minus the probability of dying.

$$P(\mathbf{x} \in Z(\ell)) = 1 - P(\mathbf{x} \notin Z(\ell)) \tag{23}$$

As the sampling of the parameters during intialization is independent among layers, the probabiliity of not dying in any of the layers of the network is the multiplication of the probabilities of not dying in each of the layers.

$$P(\mathbf{x} \notin Z(F)) = \prod_{i=1}^{D}(1 - P(\mathbf{x} \notin Z(\ell_i))) \tag{24}$$

The probability of a point to be dead for a network is equal to the probability of being dead at least in one layer, see 7. This is equivalent to 1 minus the probability of not dying in any of the layers.

$$P(\mathbf{x} \in Z(F)) = 1 - P(\mathbf{x} \notin Z(F)) \tag{25}$$

Plugging Equation 24 into Equation 25 results in Equation 22. □

**Lemma A.6** (The probability of a point to be dead for a layer increases with its depth). *Let $F : \mathbb{R}^{n_1} \to \mathbb{R}^{n_D}$ be a feed-forward neural network composed by a collection of $D \in \mathbb{N}$ layer functions $\ell : \mathbb{R}^{n_{k-1}} \to \mathbb{R}^{n_k}$ with $k \in 1, \ldots, D$, we denote $D$ as the* depth *of the network. Let the network be initialized following a random distribution. Then, given a point $\mathbf{x} \in \mathbb{R}^{n_1}$ we say that the probability of the point dying as it traverses the network (Recall Definition 7) increases with $D$.*

$$\lim_{D \to \infty} P(\mathbf{x} \in Z(F)) = 1 \tag{26}$$

*Proof.* Since the hyper-planes of the units split the space in two half-spaces, the probability of point not being dead is always lower than 1.

$$P(\mathbf{x} \notin Z(\ell)) < 1 \tag{27}$$

Therefore, the limit of the probability of $\mathbf{x}$ not dying in any of the layers when $D$ tends to infinity is zero, recall Equation 24.

$$\lim_{D \to \infty} P(\prod_{i=1}^{D}(1 - P(\mathbf{x} \in Z(\ell_i)))) = 0 \tag{28}$$

Which when plugged into Equation 22 leads to Equation 26.

□

A.3   ON THE OPTIMIZATION OF THE CONSTRAINTS

In this section we show how to differentiate the Separation Constraints with regards to the weights. Notice that under our formulation, we are examining the set diameter of the preactivation of a dataset $X$. That is, given a unit $u$ with vector parameters $\mathbf{w} \in \mathbb{R}^n$ and $b \in \mathbb{R}$, its preactivation is given by the function

$$\mathbf{z} = p(\mathbf{x}) = \mathbf{w} \cdot \mathbf{x} + b \tag{29}$$

given a finite discrete set $\mathbb{X} \subset \mathbb{R}^n$ ( a compact set of $\mathbb{R}^n$), we can define the extreme values of $\mathbb{X}$ with regards to $u$ as

$$
\begin{aligned}
z_{min}^u &= \min_{\mathbf{x} \in \mathbb{X}} p(\mathbf{x}) \\
z_{max}^u &= \max_{\mathbf{x} \in \mathbb{X}} p(\mathbf{x})
\end{aligned}
\tag{30}
$$

on a similar argument, given a layer $\ell$ with units $u_1, u_2, \ldots, u_{n_k}$ with pre-activations denoted by $p_1, p_2, \ldots, p_{n_k}$, we can define the extreme values of layer $\ell$ provided with a fixed $\mathbf{x} \in X$ as

$$
\begin{aligned}
z_{min}^p &= \min_{j=1 \ldots, n_k} p_j(\mathbf{x}) \\
z_{max}^p &= \max_{j=1 \ldots, n_k} p_j(\mathbf{x})
\end{aligned}
\tag{31}
$$

notice that since $\mathbb{X}$ is a finite discrete set of $\mathbb{R}^{n_k}$, $z_{min}^\star, z_{max}^\star$ exist (where $\star$ stands for $u$ or $p$). This fact is instrumental in our Equations (10) to (12). Since $\mathbb{X}$ is bounded, and we are working in the real number system, we can find numbers $\xi_+^\star, \xi_-^\star$ such that

$$
\begin{aligned}
z_{min}^\star &\leq -1 + \xi_-^\star \\
z_{max}^\star &\geq 1 - \xi_+^\star
\end{aligned}
\tag{32}
$$

notice that the parameters $\mathbf{w} \in \mathbb{R}^{n_k}$ and $b$ are free in the definition of $z^u$, while $\mathbf{x}$ is free in the definition of $z^p$. In this sense, for $\xi_\pm^p$ we can find $\mathbf{w}_\pm^p \in \mathbb{R}^{n_k}$ and $b_\pm^p \in \mathbb{R}$ such that

$$
\begin{aligned}
\xi_+^p &= \max(1 - (\mathbf{w}_+ \cdot \mathbf{x} + b_+), 0) \\
\xi_-^p &= \max(1 + (\mathbf{w}_- \cdot \mathbf{x} + b_-), 0)
\end{aligned}
\tag{33}
$$

that constraints all possible configurations of parameters in units of $\ell$. Naturally, $\xi_\pm$ depend on $\mathbf{w}$ and $b$. This is at the heart of the constraint formulation: choosing $\mathbf{w}$ and $b$ so that $z_{min} < 0$ and $z_{max} > 0$.

Meanwhile, given a fixed selection of $\mathbf{w}$ and $b$, since $\mathbb{X} \subset \mathbb{R}^{n_k}$ is discrete and bounded, we must have that there exist $\mathbf{x}_+$ and $\mathbf{x}_-$ points of $\mathbb{X}$ such that

$$
\begin{aligned}
\xi_-^u &\leq \max(1 - (\mathbf{w} \cdot \mathbf{x}_- + b), 0) \\
\xi_+^u &\leq \max(1 + (\mathbf{w} \cdot \mathbf{x}_+ + b, 0)
\end{aligned}
\tag{34}
$$

We wish to stress a geometric intuition, beyond calculations. We seek a combination of $\mathbf{w}$ and $b$ so that they 'cut' through $\mathbb{X}$ in the case of Sep-U, noticing that the preactivation $p$ defines a signed distance from the separating hyperplane $p(\mathbf{x}) = 0$.

The same argument is used in the field of Support Vector Machines, as cited in our references, particularly Vapnik & Cortes' Vector Support Networks from 1995 and Boyd & Vandenberghes' Convex Analysis from 2004.

A.4   ON THE RELATION BETWEEN CONSTRAINTS AND PARAMETER UPDATE

Notice that up to Equation 12, we have only stated constraints upon the slack variables $\xi_\pm$ (that involve the dataset and parameters $\mathbf{w}$ and $b$). In Equation 13 we introduce the objective function to optimize the values of $\xi_\pm$. Namely, the sum of all $\xi$.

Since we have different indications of the $\xi$, we must define different objective functions for unit-wise constraints or $g_U$ of Equation 13 and point-wise constraints or $g_P$ from Equation 15. We presented the construct for unit-based constraints (transit between Equations 12 and 13) and extended the notation for point-wise constraints on Equation 15.

If we wish to differentiate with regards to $\mathbf{w}$ and $b$ equations describing the relation between $\mathbf{x}$ and $\xi$ unit-wise or point-wise, we must notice that $\nabla_w \xi_{\pm}^u$ as stated, must be defined by parts:

$$\nabla_w \xi_-^u = \begin{cases} \mathbf{x}_-, & p(\mathbf{x}_-) > -1 \\ 0, & \text{otherwise} \end{cases} \tag{35}$$

$$\frac{\partial \xi_+^u}{\partial b} = \begin{cases} 1, & p(\mathbf{x}_-) > -1 \\ 0, & \text{otherwise} \end{cases} \tag{36}$$

while for $\xi_+^u$:

$$\nabla_w \xi_+^u = \begin{cases} -\mathbf{x}_+, & p(\mathbf{x}_+) < 1 \\ 0, & \text{otherwise} \end{cases} \tag{37}$$

$$\frac{\partial \xi_+^u}{\partial b} = \begin{cases} -1, & p(\mathbf{x}_+) < 1 \\ 0, & \text{otherwise} \end{cases} \tag{38}$$

while under the unit-based formulation $\mathbf{x}_{\pm}$ is fixed by the geometry of $\mathbb{X}$, for the point-wise formulation, we must have

$$\nabla_{w_-} \xi_-^p = \begin{cases} \mathbf{x}, & p(\mathbf{x}) > -1 \\ 0, & \text{otherwise} \end{cases} \tag{39}$$

$$\frac{\partial \xi_-^p}{\partial b} = \begin{cases} 1, & p(\mathbf{x}) > -1 \\ 0, & \text{otherwise} \end{cases} \tag{40}$$

and for $\xi_+^p$:

$$\nabla_{w_+} \xi_+^p = \begin{cases} -\mathbf{x}, & p(\mathbf{x}) < 1 \\ 0, & \text{otherwise} \end{cases} \tag{41}$$

$$\frac{\partial \xi_+^p}{\partial b} = \begin{cases} -1, & p(\mathbf{x}) < 1 \\ 0, & \text{otherwise} \end{cases} \tag{42}$$

that is parallel to the loss gradient $\nabla_w L$.

### A.5 Effect of the separation on the internal representations

In this section we explore the effect of the proposed constraints in the activation of the network. For that task we train all methods fixing the architecture at 50 layers of 4 units each. We use the `MOONS` dataset for easy visualization, in the same configuration than Section 4(sampling 100 points, 85 for training and 15 for validation) .

**Experimental setting:** We compare the combination of `Sep-U`, `Sep-P` and `Sep-UP` to feed-fordward `ReLU` networks (Glorot et al., 2011) and Batch Normalization as described in Ioffe & Szegedy (2015) using the same architecture.

**Training Parameters:** All the networks used were optimized using Adam (Kingma & Ba, 2014). We used a learning rate of $0.01$ for 1000 epochs and a batch size of 85. We used $\lambda = 10^{-4}$ when required. As initialization scheme, we used Glorot uniform from Glorot & Bengio (2010). We also test the use of the Separation Constraints alone, without categorical crossentropy, to assess its effect by itself. Our experiments were conducted using `Keras` (Chollet et al., 2015) and `TensorFlow` (Abadi et al., 2015), fixing the random seed to an arbitrary value of 10.

Table 1 summarizes the results. Figures 6, 7, 11, 10 and 8 show the activation plots for each of the methods tested.

|         | Accuracy | | Loss | |
|---------|----------|------|--------|--------|
|         | Train | Val. | Train | Val. |
| ReLU | 0.5176 | 0.4 | 0.6925 | 0.6938 |
| ReLU + BN | 0.8117 | 0.6 | 0.6331 | 0.6636 |
| Sep-P | 0.9294 | 0.8000 | 0.1765 | 0.6476 |
| Sep-U | 0.9058 | 0.8000 | 0.4161 | 1.5228 |
| Sep-UP | 0.9882 | 0.9333 | 0.6988 | 1.0810 |

Table 1: Maximal performance experiment using the `MOONS` dataset. From left to right, accuracy and loss (for *train* and *validation* sets) for `ReLU`, `ReLU + BN`, and `Sep-Cons` in all its variants.

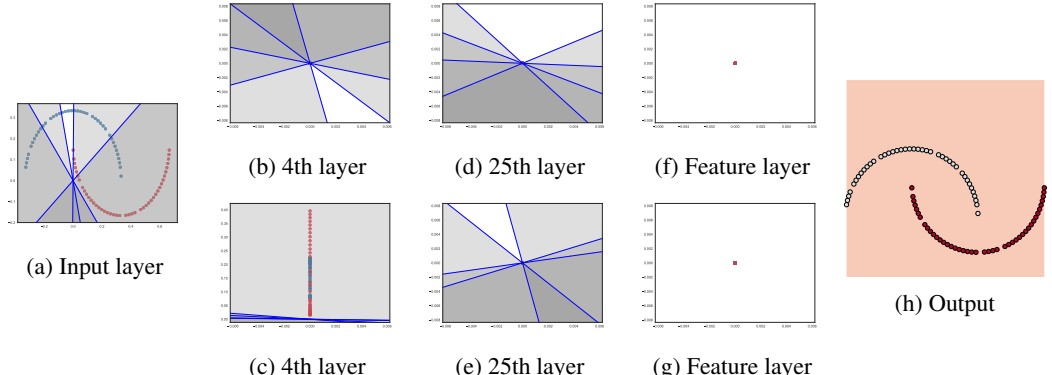

(a) Input layer    (b) 4th layer    (d) 25th layer    (f) Feature layer    (h) Output

(c) 4th layer    (e) 25th layer    (g) Feature layer

Figure 6: Data transformed across a 50x4 `ReLU` classification network. Notice how the the dataset is progressively mapped to zero as it traverses the network. This renders the output layer unable to solve the problem.

In terms of accuracy `ReLU` reaches a trivial accuracy of $0.51$ while `ReLU + BN` reaches $0.60$, both architectures fail to solve the problem, see Table 1. When comparing the results presented in Figures 6 and 7, we find that at the fourth layer `ReLU` has collapsed the points of the dataset over a line (parts 6b and 6c), while `ReLU + BN` still manages to warp the dataset. However, both methods at layer 25th have *collapsed* the dataset into a small number of points (see Figures 7d, 7e for `ReLU + BN` and Figures 6d and 6e for `ReLU`) that is then *pushed* to the point zero for `ReLU` (as shown in the feature layer: 6f and 6g), while `ReLU + BN` collapses to few points (Figures 7f and 7g).

This collapse is congruent with our expectations, displaying the *born-dead network* condition from Lu et al. (2019). Note how layer after layer its effect is worse, as we predicted in A.2. In addition, since the gradient is only back-propagated through the points lying on the upper sets of units, geometric collapse also stops learning. Notice that the standarization and the affine transformation (dependent on $\gamma$ and $\beta$) from `ReLU + BN` are not able to prevent from killing the dataset. Thus, training thin and deep neural networks is difficult with `ReLU` or `ReLU + BN`.

On the extent of our experimentation `ReLU` networks featuring separation constraints are able to solve the problem with higher degrees of accuracy than `ReLU` and `ReLU + BN` (see the figures in Table 1). Moreover, Figures 8h, 11h and 10h), show that the separation function behaves *intuitively* in the sense of Hauser & Ray (2017).

The internal representations not only are non-trivial (as in `ReLU + BN`) but also preserve geometrical structures like shape and connectivity, as shown in Figures 11d, 10d or 8e. Indeed, Figures 11c, 8c and 10c showcase how at the 4th layers a much a solution is already found. This proves that the gradient of the main loss is back-propagated to the input, unlike `ReLU` and `ReLU + BN` (Figures 6b, 6b, 7b, 7c).

The constraints enforce richer representations when used without main loss (cross-entropy),

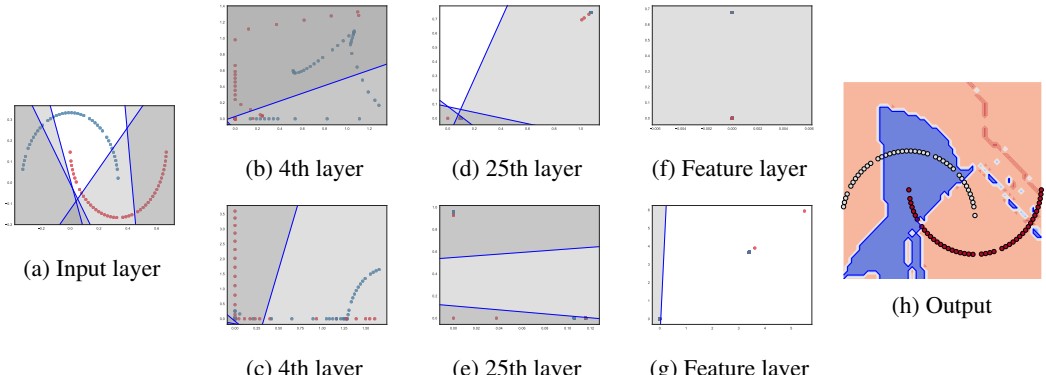

Figure 7: Data transformed across a 50x4 `ReLU + BN` network trained using `Adam` with a learning rate of 0.01 in the `MOONS` dataset for 1000 epochs. The dataset is collapsed in few points at the feature layer. As the gradient cannot be backpropagated across the truncation after the affine transform of $\gamma$ and $\beta$ despite the standarization, it fails in the same manner than `ReLU` only that with non-zero activations. This results in *topological mixing* of the datasets. Therefore, the representational capability of the network is hindered to such extent that the resulting output, although non-trivial, is totally arbitrary.

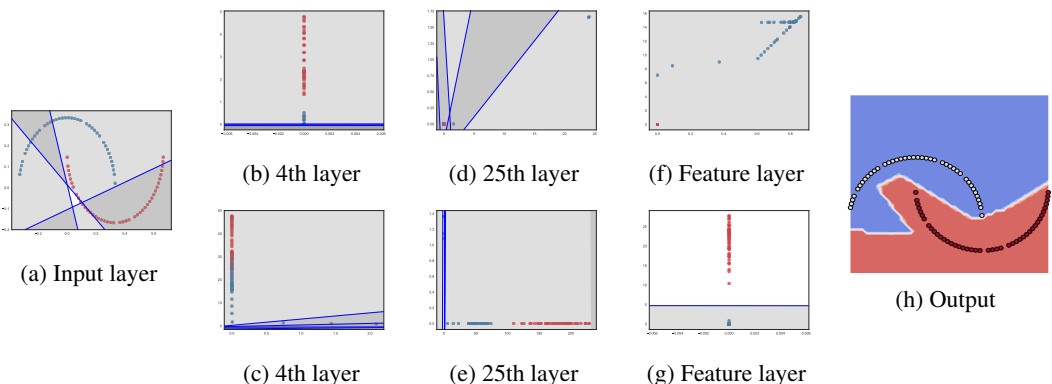

Figure 8: Data transformed across a 50x4 `Sep-UP` network trained using `Adam` with a learning rate of 0.01 in the `MOONS` dataset for 1000 epochs. The network displays internal representations without collapsing the dataset like `Sep-P` and retaining discriminative power like `Sep-U`.

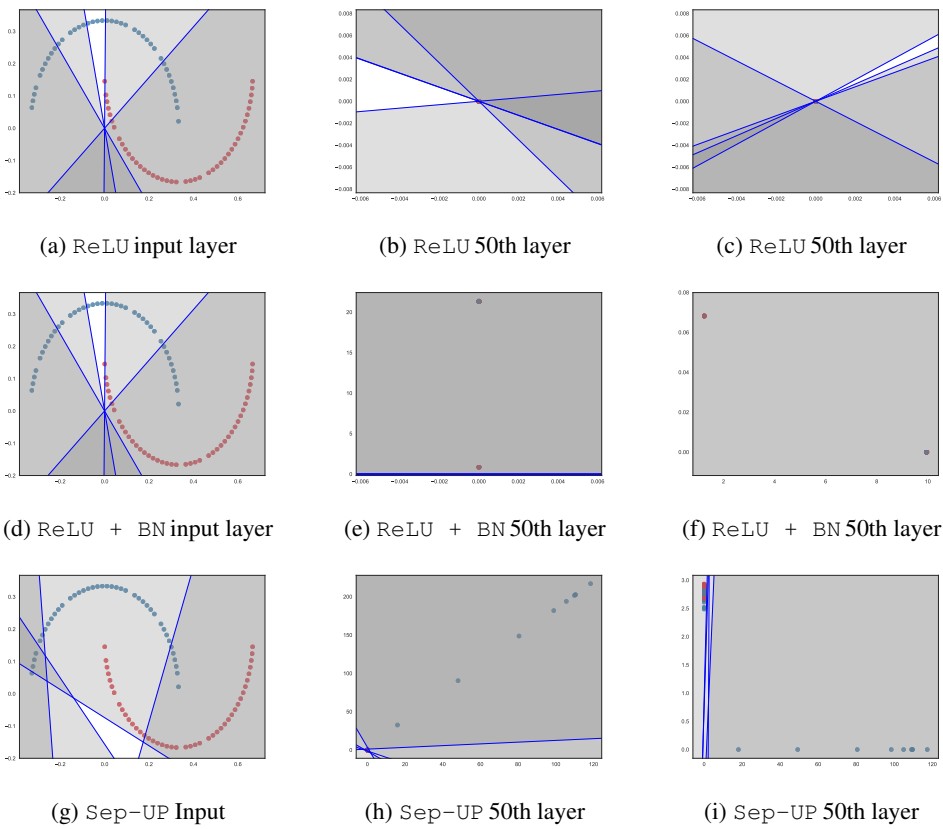

(a) `ReLU` input layer      (b) `ReLU` 50th layer      (c) `ReLU` 50th layer

(d) `ReLU + BN` input layer    (e) `ReLU + BN` 50th layer    (f) `ReLU + BN` 50th layer

(g) `Sep-UP` Input      (h) `Sep-UP` 50th layer      (i) `Sep-UP` 50th layer

Figure 9: Data transformed across a 50x4 network with no main loss (cross-entropy) with constraints `Sep-L` and `Sep-UP` , versus `ReLU` and `ReLU + BN` . Notice how effectively `ReLU` and `ReLU + BN` collapse the dataset into few points whereas `Sep-UP` force the network to learn representations that preserve geometrical structure useful for back-propagation.

see Figure 9. whereas the representation reached by `Sep-UP` is more complex, yet preserves connectivity. Furthermore, both avoid mapping the entire dataset to few values as observed with `ReLU` (Figures 9e and 9f) and `ReLU + BN` (Figures 9b and 9c).

The only exception is `Sep-U` that apparently solves the problem at layer 25th on Figures 11d and 11e, yet collapses the dataset in two points at the feature layer (Figures 11f and 11f). Our intuition is is that, although `Sep-U` prevents dead units, it cannot prevent points from falling into the intersection lower set of the units of a layer, resulting in a *dead point*). The `Sep-P` separation constraint was designed precisely to prevent the presence of dead points. However, `Sep-P` allows affine and dead units (as shown in Figures 10d and 10e), so the the decision function reached becomes too linear (see Figure 10h).

Therefore, if we combine both `Sep-U` and `Sep-P` we should have best of both worlds, as we say in Section A.1. Figure 8, shows how `Sep-UP` is able to separate both classes perfectly (recall the 0.93 in accuracy from Table 1). Furthermore `Sep-UP` , produces a feature layer that does not concentrates the dataset in two points like `Sep-U` , nor distributes the dataset linearly like `Sep-P` .

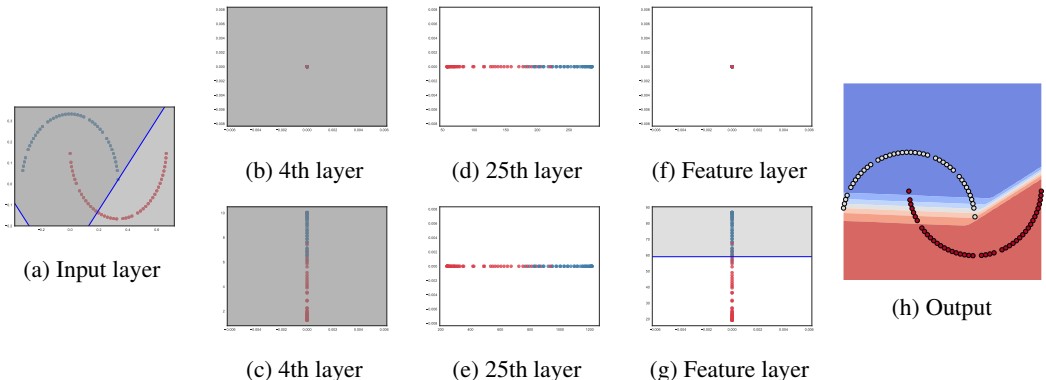

Figure 10: Data transformed across a 50x4 `Sep-P` network trained using `Adam` with a learning rate of 0.01 in the `MOONS` dataset for 1000 epochs. The network displays a richer internal representation without collapsing the dataset like `Sep-U` or `ReLU + BN`. However, plenty of dead units appear since they are not penalized, causing underfitting.

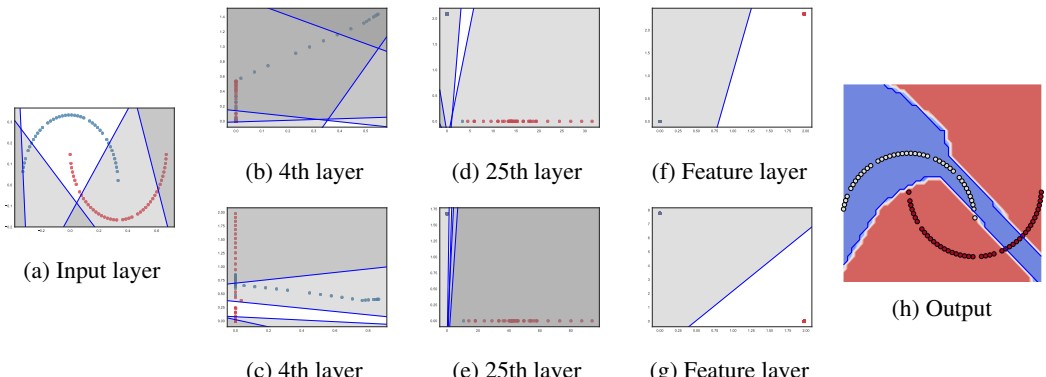

Figure 11: Data transformed across a 50x4 `Sep-U` network trained using `Adam` with a learning rate of 0.01 in the `MOONS` dataset for 1000 epochs. Notice how dead units have been reduced. The interal representations are much richer than `ReLU` or `ReLU + BN`. Despite collapsing the dataset in two points at the feature layer, the classification performed in the output layer is approximately correct. We conjecture that this is due the dead point addressed with `Sep-P`.

A.6   RELATION BETWEEN SEPARATION CONSTRAINTS AND MAIN LOSS

In this section we explore the effect of the proposed constraints in the convergence of the network during optimization. We train a network of 50 layers of 4 units each using `Sep-UP`. We use the `MOONS` dataset, in the same configuration than Section 4 (sampling 100 points, 85 for training and 15 for validation).

**Experimental setting:** We analyze the interaction between of the main loss (cross-entropy) and the loss arising from the Separation Constraints. We plot them, together with training accuracy in Figure 12. Below we attach the decision boundaries at the most relevant epochs.

**Training Parameters:** The networks wes optimized using Adam (Kingma & Ba, 2014). We used a learning rate of 0.01 for 17000 epochs and a batch size of 85. We used $\lambda = 10^{-4}$. As initialization scheme, we used Glorot uniform from Glorot & Bengio (2010). Our experiments were conducted using `Keras` (Chollet et al., 2015) and `TensorFlow` (Abadi et al., 2015), fixing the random seed to an arbitrary value of 10.

We see at Figure 12 how the use of the separation constraints enables back-propagation, and thus cross-entropy minimization. According to it, the training begins minimizing the constraint loss until a certain level is reached (approximately $10^{-2}$ circa epoch 250). During this phase, the cross-entropy loss stays constant. Only when this critical point in the constraint loss is reached, back-propagation is restored and the accuracy starts climbing up to 100%. However, using an this additional loss (from the separation constraints) induces *aggressive* transient states (e.g. instabilities) during training (approximately around epochs 6200, 11883 and 16432). We see also how after each of the transient states, the network finds a new solution different from earlier ones. Moreover, those solutions show an increasing level of complexity, in the form of smoother and more rouded boundaries. For instance, the first solution (Figure 12f) is linear and the second is composed of few lines (Figure 12h, whereas the last and second to last (Figure 12ai and 12ao) are much more curved.

We conjecture that this behaviour must be due dead units and points being revived during training. As a dead unit or point comes back from being dead, the weight configuration set by the Separation Constraints is likely to be unrelated to cross-entropy. That causes a break down in the convergence that we can see around epochs 6200, 11883 and 16432. As back-propagation is now working for those points or units, their error is eventually corrected by normal cross-entropy minimization and the network converges to a solution again. However, as now there is an additional working unit or point involved, the network has greater representational expressiveness, which results in a more complex decision boundary.

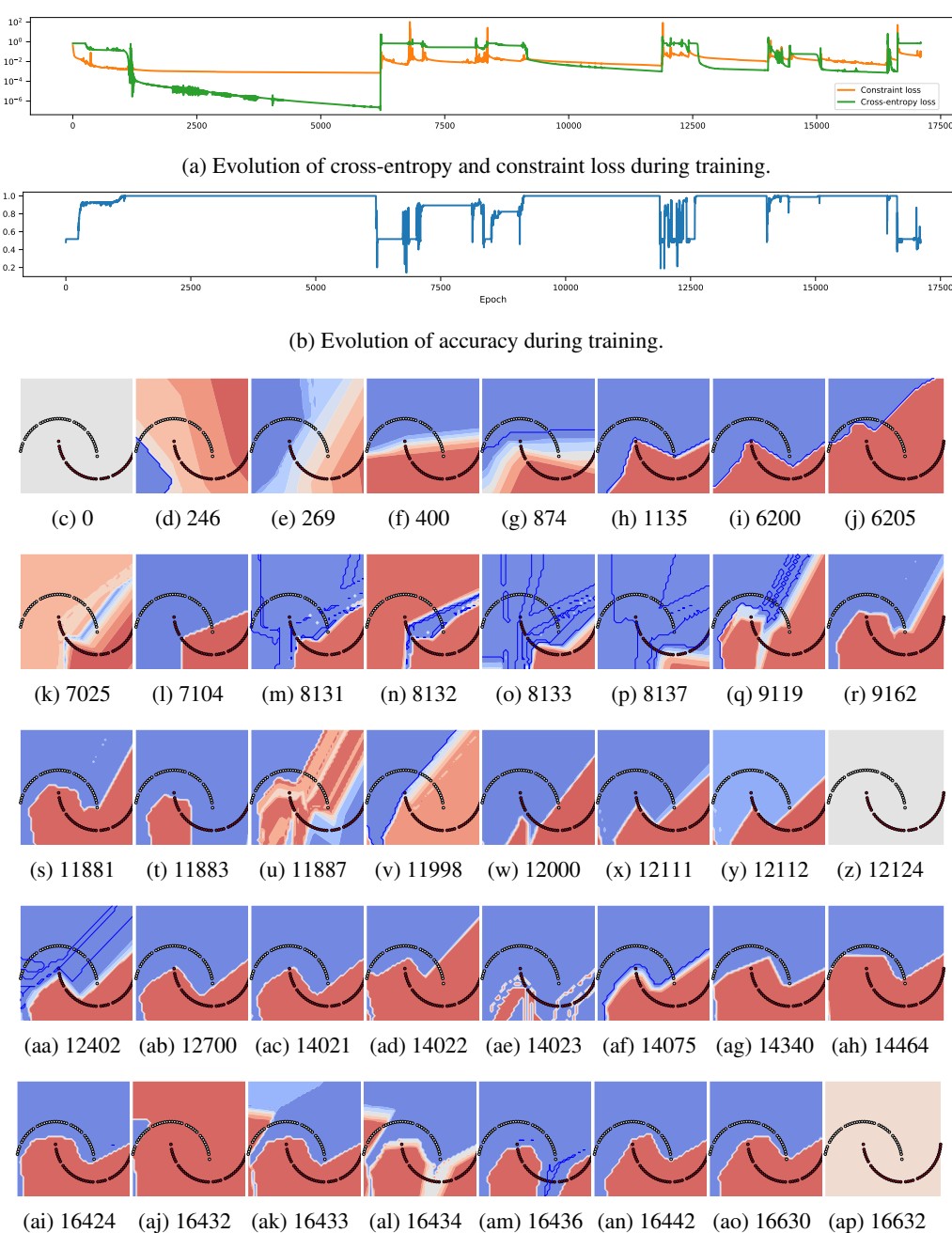

Figure 12: Evolution of training throughout epochs (cross-entropy, constraint loss, accuracy, and decision surface) in 50x4 network trained on the MOONS dataset. We used Adam with a lr of $0.01$ for $17000$ epochs and a batch size of $85$ and Glorot uniform. We used $\lambda = 10^{-4}$. Figure 12a shows both cross-entropy and constraint loss for each epoch of the training phase in the horizontal axis, whereas Figure 12b does the same for training accuracy. The rest of subfigures show the decision surface at the indicated epoch. We see how learning only starts after the constraint loss has been minimized below approximately $10^{-2}$, circa epoch 250. After that, back-propagation is restored and cross-entropy falls quickly, reaching a solution at epoch 1135. However, around epoch 6205 the training breaks down and the next epochs are spent in a transient state, until the geometry of the boundary changes and the network converges to a different solution (note how the solution found at 1135 is composed of 5 lines, whereas 9162 consists of 7). This process occurs several times (6200, 11883 and 16432), resulting in increasingly complex solutions (notice how solution at 16630 is much more rounded and smoother). We conjecture that this due to dead units and points recovered during training by the Separation Constraint augment the capacity of the network.

