# OpenReview forum: "Superseding Model Scaling by Penalizing Dead Units and Points with Separation Constraints"
_ICLR.cc/2020/Conference — Reject_

### Official Review · AnonReviewer2 · 2019-10-19
**Official Blind Review #2**

**Rating:** 3

**Review:**

In this paper, a problem of training DNNs having dead points and neurons was addressed. The problem was posed as a constraint optimization problem. The constraints were augmented to the loss functions of DNNs as a regularizer. The slack variables of the constrained loss functions were optimized together with parameters of DNNs to train DNNs avoiding dead points and neurons.

The paper was written well in general. However, the work is incomplete in terms of explanation of the details of the proposed method, and their experimental analyses. For instance:

- In the code, you implemented the constraints in a ReLU.  In this case, could you please compare the proposed SeparatingReLU with the other variation of ReLU such as parametric ReLUs etc.

- Could you please analyze convergence of parameters of DNNs together with slack variables during training?

- Do you apply additional constraints on slack variables to control their scale?

- How do the proposed methods scale with larger datasets and networks? Please provide more detailed analyses for training larger/wider DNNs with different structures (such as VGG and ResNets) on larger datasets, such as at least Cifar 100 and ImageNet.

After the discussion:

I checked the response of the authors and comments of the other reviewers.

The authors consider analyses of various crucial properties of the methods in comparison with the other works as the future work. There are also some statements need to be verified and explained more precisely, such as regarding generalization properties and the relationship between network complexity, constraints and regularization. Therefore, I suggest authors to improve the related statements, and provide more detailed analyses for verification of the proposed results in the following submissions.

**Experience Assessment:**

I have published in this field for several years.

**Review Assessment: Checking Correctness Of Derivations And Theory:**

I carefully checked the derivations and theory.

**Review Assessment: Checking Correctness Of Experiments:**

I carefully checked the experiments.

**Review Assessment: Thoroughness In Paper Reading:**

I read the paper at least twice and used my best judgement in assessing the paper.

---

### Official Review · AnonReviewer3 · 2019-10-24
**Official Blind Review #3**

**Rating:** 3

**Review:**

This papers suggests new penalties to avoid the apparition of dead units and minimize the existence of dead points.

Decision

I vote to reject this paper because the formulation of the penalties lacks clarity and because the experiments are incomplete and likely provide misleading results.

Justification

    Formulation of the penalty

It is unclear to me how the slack variables xi depends on the weights, and therefore how they can be minimized by adjusting the weights using a penalty. I believe this should be clarified between equations (12) and (13).

The optimizer used is Adam instead of a plain SGD. How is the penalty integrated in Adam? Is it left out as it is best to do for weight decay or is it integrated in the adaptive estimations? I believe a plain SGD would be better justified when the goal is to understand an improvement to the flow of information in the network.

    Experiments

The experimental setup have many problems.
1) The protocol is creating a bias
2) The comparisons of zero initializations are indirect
3) The tasks chosen cannot corroborate that the penalty is helping information flow deeper without harming generalisation.

1) Protocol
Results may be misleading both because learning rates are not adjusted, which can cause deeper models to diverge while they could still be trained and because number of epochs is limited instead of letting the models train until convergence. I note that plots comparing train and validation accuracy in Figure 1 shows better results on the validation set than on the training set for ReLU and ReLU + BN. This suggests indeed that the optimization diverged, breaking the training accuracy while leaving the validation accuracy close to random. It is difficult however to evaluate the color-maps without an axis for the colors, so my assumption that dark purple is close to random accuracy may be wrong. A related note, description of Figure 4 in section 4.2 discuss about the achieved maximum performance at depth 60 and width 25. I could not read the same, to me it seems to be achieved right at depth 1, width 3.

2) Comparisons of zero initializations
The experiments on the zero initialization scheme is an interesting investigation, but it is incomplete as there is no direct comparison with training without Sep-UP. The Annealed Dropout is important to allow training with zero initialization, and may well be more important than Sep-UP. We cannot measure their respective importance without a direct comparison as in Figure 1.

3) Too simple tasks
The Figure 3 points to an important issue of the paper. Generally, the depth of networks is shown to provide higher capacity which leads to improved accuracy when not overfitting. In the experimental setup of this paper, the tasks and architectures do not improve with depth, or more precisely the improvement in capacity (training accuracy is maximal at depth 10) is associated with overfitting. If empirical results show that the contributed penalty helps improving the flow of information, it should also be shown that it is not at the cost of better generalisation otherwise there is no point in having greater depth. The chosen experimental setup does not allow this however. Even though it does fair better at generalization than the others, they all suffer from worst generalisation with depth. More difficult tasks may be better suited for these experiments. I am not recommending to use very large datasets such as ImageNet, difficult synthetic problems could be sufficient. It is necessary however that depth can be shown to be beneficial for baselines, so that we can confirm that training with the penalty is not degrading the beneficial effect of depth.

Other comments

The problem of dead points should be shown empirically since the contributions of this paper are supported empirically.

Fourth paragraph of section 4.1 mentions results using convolutional layers, but it is not clear which of the provided results are on fully connected or convolutional networks.

Minor comments

In equation (1), \hat{e}_j is not specified. My guess is that represents a basis, but I don't see why this basis necessary in the equation.
In equation (5), the notation is limited to a single layer, with x being the input of the layer. I believe this should be generalized with a composition operator as in equation (7).
There seems to be a mistake in equation (8). The inclusion in X on the right is useless since the set on the left is an intersection with X.
Section 3.1, third paragraph: It is *easily* to see [...]
Section 3.2, first paragraph: [...] slack variables for *each each* point on the batch.
Section 4.1, first paragraph: [...] (see Figures 1b and *1b*). (should be e)

Edit:

I still believe this paper lacks empirical evidence that the proposed loss would help train deep models without harming performance on tasks where depth matters. In light of all the corrections and clarifications, I upgrade my vote to weak reject.

**Experience Assessment:**

I have read many papers in this area.

**Review Assessment: Checking Correctness Of Derivations And Theory:**

I carefully checked the derivations and theory.

**Review Assessment: Checking Correctness Of Experiments:**

I carefully checked the experiments.

**Review Assessment: Thoroughness In Paper Reading:**

I read the paper thoroughly.

---

### Official Review · AnonReviewer1 · 2019-10-31
**Official Blind Review #1**

**Rating:** 3

**Review:**

This paper proposed two constrains to tackle the problems of dead points and dead neurons. In the toy datasets,  these two problems are successfully solved. In addition, the network weights can be initialized naively with the proposed method.

Pros:
- The idea sounds interesting. Dead points and dead neurons are solved in the toy examples.
- Networks can be initialized with zeros with the proposed idea.

Cons:
- The experiments are not sufficient to validate the arguments.  MNIST and CIFAR-10 datasets are toy datasets. As the memory complexity is reasonable, Experiments should be done on ImageNet.
- Why aren't the constrains 12 and 13 in the final objective 16? Am I missing something?
- Too many typos.

minors:
e hat is not defined in equation 1.

**Experience Assessment:**

I do not know much about this area.

**Review Assessment: Checking Correctness Of Derivations And Theory:**

I assessed the sensibility of the derivations and theory.

**Review Assessment: Checking Correctness Of Experiments:**

I assessed the sensibility of the experiments.

**Review Assessment: Thoroughness In Paper Reading:**

I read the paper at least twice and used my best judgement in assessing the paper.

---

### Author Response · Authors · 2019-11-12
**3. SPECIFIC COMMENTS THAT WE WISH TO ADDRESS**

3. SPECIFIC COMMENTS THAT WE WISH TO ADDRESS

We wish to address some other comments from Reviewer #2:

        - Do you apply additional constraints on slack variables to control their scale?

No.

        The constraints were augmented to the loss functions of DNNs as a regularizer.

The separation constraints are not a regularizer. A regularizer is a term in the objective function that penalizes additional complexity. The constraint formulation intends exactly the opposite. It forces the network to become more complex, by forcing each unit to perform a non-linear (piece-wise defined) operation.

         - In the code, you implemented the constraints in a ReLU. In this case, could you
         please compare the proposed SeparatingReLU with the other variation of ReLU
         such as parametric ReLUs etc

We believe that testing other variations of ReLU is out of the scope of the paper, which is focused on training deeper networks without using the layer width increase technique (a.k.a. width scaling). However, we plan to do so in the future, as we say in the Conclusions. We already preliminary tested tanh and it improved from 0.5 to 0.74 in MOONS.


REFERENCES

Gao Huang, Zhuang Liu, and Kilian Q. Weinberger.   Densely connected convolutional networks.
    CoRR, abs/1608.06993, 2016. URL http://arxiv.org/abs/1608.06993.

Lu  Lu,  Yeonjong  Shin,  Yanhui  Su,  and  George  Em  Karniadakis.   Dying  relu  and  initialization:
    Theory and numerical examples. arXiv preprint arXiv:1903.06733, 2019.

Mingxing Tan and Quoc V. Le.   Efficientnet:  Rethinking model scaling for convolutional neural networks.
    CoRR, abs/1905.11946, 2019. URL http://arxiv.org/abs/1905.11946.

Ligeng Zhu,  Ruizhi Deng,  Zhiwei Deng,  Greg Mori,  and Ping Tan.   Sparsely connected convolutional networks.
    CoRR, abs/1801.05895, 2018.   URL http://arxiv.org/abs/1801.05895

---

### Author Response · Authors · 2019-11-12
**2.8 USE OF CONVOLUTIONAL LAYERS, 2.9 ON THE CONVERGENCE OF THE PROPOSED METHOD, 2.10 ABOUT TESTING THE SEPARATION CONSTRAINTS ON VGG OR RESNET**

2.8 USE OF CONVOLUTIONAL LAYERS

Reviewer #3 states

        Fourth paragraph of section 4.1 mentions results using convolutional layers, but it
        is not clear which of the provided results are on fully connected or convolutional
        networks.

Convolutional layers are used with images from MNIST and CIFAR-10 datasets. We have clarified the statement.

2.9 ON THE CONVERGENCE OF THE PROPOSED METHOD

Reviewer #2 asks

 - Could you please analyze convergence of parameters of DNNs together with slack variables during training?

Similarly to our resolve concerning other ReLU variations, we claim that it deserved its own space (its own publication). However, as we mention in the Conclusions, as we experimented with the constraint formulation, we found an interesting property: Initially, the network converges to a solution nicely, but such favorable configuration is not maintained. After some epochs, the training process suffers a break down. Then, after spending some epochs in a transient state, the network converges to a solution again.

We conjecture that is because as the training progresses the Separation Constraints devote themselves to raise dead units back to live (as cinematic as it sounds). However, the 'reborn' units have not been optimized with regards to the main loss (as they did not have access to cross-entropy backpropagation when they were dead). As they are optimized the network 'struggles' with them, so that it converges to a solution again and stability is reached for some epochs, until the same phenomenon happens again.

The most interesting part is that if we plot the decision surface (which we can do, another reason for choosing the MOONS dataset) we see that the solutions found after each breakdown is increasingly more complex (graphically-wise): the first configurations are rough (composed of three or four lines), whereas the latter configurations reached are smoother and follow a curved boundary that fits better the shape of MOONS.

Since this is far beyond the scope (and the space) of the paper we decided to leave it for future research. However, we have recovered a convergence plot and put it on Appendix Section A.4, in deference to  Reviewer \#2's interest.

2.10  ABOUT TESTING THE SEPARATION CONSTRAINTS ON VGG OR RESNET

Reviewer #2 asks

        How do the proposed methods scale with larger datasets and networks?  Please
        provide  more  detailed  analyses  for  training  larger/wider  DNNs  with  different
        structures (such as VGG and ResNets) on larger datasets, such as at least Cifar
        100 and ImageNet.

We believe that applying the constraint formulation to specifically tailored architectures (such as VGG or ResNets) has no additional effects performance-wise, precisely because they are engineered to solve the problem (of dead units/points) by other means.

In the case of VGG (and almost all current production grade architectures), the same effect of the separation constraints is achieved through the means of the width increase strategy (as stated in our introduction). Therefore, applying the separation constraints is likely to show no effect, as back-propagation is already working thanks to the additional (alive) units found by increasing width.

In the case of ResNet, in addition to the width increase, the introduction of additional connections helps back-propagating the gradient (towards the input). However, the summation done in ResNet (an integral part of its aggregation mechanism) has been reported as a source of saturation as the networks grows (Zhu et al., 2018, Section 3.1).

A priori we do not find enough analytical grounds to suggest that it would make a difference performance-wise. However, testing in thinned versions of VGG and ResNets could be interesting in the future.

---

### Author Response · Authors · 2019-11-12
**2.6 ON OUR TOO SIMPLE TASKS AND 2.7 EMPIRICAL EVIDENCE OF DEAD POINTS**

2.6  ON OUR TOO SIMPLE TASKS

We have elaborated on our choice of datasets (and others) used in second section of this article, so that will only discuss here reviewers' comments about the role of depth and overfitting. We will start with Reviewer \#3.

        The Figure 3 points to an important issue of the paper.  Generally, the depth of
        networks is shown to provide higher capacity which leads to improved accuracy
        when not overfitting.  In the experimental setup of this paper, the tasks and archi-
        tectures do not improve with depth, or more precisely the improvement in capacity
       (training accuracy is maximal at depth 10) is associated with overfitting

We fail to see the connection of this phenomenon with overfitting. In our understanding, overfitting is measured as the difference between training and validation error. In Figure 3 we notice that this difference with Sep-UP is not only maintained but even reduced (from 0.79-0.57 at layer 10, to 0.68-0.53, lowering from 0.22 to 0.15). In fact, what we conjecture is that training error is increased with depth, as a product of the occurrence of more dead units and points, as predicted.

Alleviating this problem is precisely the goal of our paper (as sated on the introduction). Furthermore, we see how Sep-UP mitigates accuracy-decrease considerably in comparison to ReLU-BN (i.e. ReLU breaks down straight-forwardly).

Reviewer #3 follows with

        If empirical results show that the contributed penalty helps improving the flow of
        information, it should also be shown that it is not at the cost of better generalisa-
        tion otherwise there is no point in having greater depth. The chosen experimental
        setup does not allow this however.  Even though it does fair better at generaliza-
        tion than the others, they all suffer from worst generalisation with depth.  Even
        though it does fair better at generalization than the others, they all suffer from
        worst generalisation with depth. More difficult tasks may be better suited for these
       experiments. I am not recommending to use very large datasets such as ImageNet,
       difficult synthetic problems could be sufficient. It is necessary however that depth
       can be shown to be beneficial for baselines, so that we can confirm that training
       with the penalty is not degrading the beneficial effect of depth.

If the constraints were harmful to generalization,  we would have verified it in all our experiments with them, not only on the ones using CIFAR-10. This suggests that the cause of this issue must be searched elsewhere. In addition, since the complexity of CIFAR-10 is much greater than MNIST, we understand that insufficient width choices (for the network) due limited computational resources are to blame for underfitting, producing a meager 0.53 accuracy at depth 40.

A relevant question, is implicitly posed here and is part of the motivation behind taming 'extra depth'. If the problem could be solved with a shallow network there is no point going deeper. However, if depth is required (e.g. for really complicated tasks), stopped back-propagation due to dead units and points will eventually appear (recall our presentation of dead units), as it is a problem intrinsic to the network, invariant to the dataset. It is in this case when when the Separation Constraints shine, as they produce better performance with the same width than their ReLU-BN counterparts, (as we show in Figures 1, 2 and 3).

2.7  EMPIRICAL EVIDENCE OF DEAD POINTS

Reviewer #3 states that

         The problem of dead points should be shown empirically since the contributions
         of this paper are supported empirically.

We provide empirical evidence in the appendix in the form of figure 5d. Notice that penalizing only dead points with Sep-P works with networks as deep as 70 layers and 40 when using the minimum width, compared to 20 and 10 respectively of ReLU.

If there were no dead points, penalizing them should produce no improvement over the ReLU baseline. In addition, in the activation plots added in the Section A.3.1 Figure 10, it is easy to see them: without penalizing dead points the boundary becomes almost linear.

Notice how the activations of ReLU-BN (Figure 7a) or Sep-U (Figure 11a) show abundance of dead points (the points lying in the white area, this is, in the negative intersection of the planes).

Additionally, we provide the reader with mathematical proof that dead points appear (asymptotically) as  networks grows deeper (check the Appendix,  Section A.2). We removed it from the submitted version, due to length constraints of the ICLR, but we have recovered it, in light of Reviewer \#3's interest on the matter.

---

### Author Response · Authors · 2019-11-12
**2.4 ABOUT OUR COLORMAPS AND 2.5 ZERO INITIALIZATION**

2.4  ABOUT OUR COLORMAPS

Reviewer #3 states that:

         It is difficult however to evaluate the color-maps without an axis for the colors, so
         my assumption that dark purple is close to random accuracy may be wrong

We have updated the article with colormaps for each Figure. However, as the caption states "The color shows the accuracy attained of each of the combinations of width and depth. This implies that light beige is an accuracy of $1$ and black is a trivial accuracy of $0.5$.". We also provide with the raw numbers in csv files within the zip folder of the code.

Reviewer #3 also comments:

         A related note, description of Figure 4 in section 4.2 discuss about the achieved maximum performance at depth 60 and
         width 25. I could not read the same, to me it seems to be achieved right at depth 1, width 3.

Both are correct. Indeed, what we state is that at depth 60, Zero initialization requires a width of 25 units, whereas Glorot needs only 2 units per layer (Figure 1) to function. We have changed the sentence clarifying it.


2.5  ABOUT OUR APPROACH ON ZERO INITIALIZATION

Reviewer \#3 states that:

        The experiments on the zero initialization scheme is an interesting investigation,
        but it is incomplete as there is no direct comparison with training without Sep-
        UP. The Annealed Dropout is important to allow training with zero initialization,
        and may well be more important than Sep-UP. We cannot measure their respective
        importance without a direct comparison as in Figure 1.

Using zero initialization in ReLU or ReLU-BN networks forces all outputs of the network to become zero. This produces a zero-gradient and stops learning. By adding Annealed Dropout it is impossible to improve on anything, as all it will do is to set some values to zero (that were already zero). Hereby, we did not believe that testing against ReLU or ReLU-BN was needed.

---

### Author Response · Authors · 2019-11-13
**2 ABOUT TECHNICAL ASPECTS OF THE CONSTRAINT FORMULATION AND OUR EXPERIMENTS: 2.1 CHOICE OF THE DATASET, 2.2 CHOICE OF THE OPTIMIZER AND 2.3 ABOUT OUR EXPERIMENT PROTOCOL**

2  ABOUT TECHNICAL ASPECTS OF THE CONSTRAINT FORMULATION AND OUR EXPERIMENTS

We begin this section discussing a shared concern of all reviewers: our choice of dataset, to follow with some remarks about our optimizer (Adam), and some comments on our experimentation protocol, graphics and zero initialization. We wish also to address particular comments on how our paper is done using 'too simple tasks', and finish with several specific comments made from the reviewers on our experiments.

2.1 CHOICE OF THE DATASET

Another complaint shared among the reviews is the choice of the dataset.  Of course, if we were proposing a novel method to get better accuracy it would be needed to test it in Imagenet (like everyone else), but it is not the case. We are solely interested in alleviating the dependence of width in depth, which is a intrinsic property of neural networks (Lu et al., 2019, Theorem 1), thus invariant to dataset.

Additionally, the use of more complex datasets, e.g. ImageNet or synthetics, would defeat the purpose of the experimentation. Since we are interested in the problems arising from dead points and units when the networks grow deeper, we need a dataset a simple as possible in order to isolate error from other sources  as underfitting or overfitting.

We are only concerned with alleviating dead points and units to enable back-propagation, nothing more. That is why we chose a very simple dataset which can be solved with three units, to show that if the network fails, it must be due to depth-induced problems.

Additonally, the complexity on the number of constraints is (theoretically) low, but we suspect that our implementation does not reuse the activations. This results in creating new matrices (in memory) that limit experimentation when using a single GTX1080Ti.

Proper optimization would address this problem, but we our expertise in writing custom kernels in Tensorflow is insufficient. Finally, our interest lies in sharing the results of the prototype presented in our work, so that others (with more experience in performing large scale experiments, and possessing  greater computational power) may profit from our work.

2.2  ON OUR CHOICE OF OPTIMIZER

Reviewer #3 affirms:

         The optimizer used is Adam instead of a plain SGD. How is the penalty integrated
         in Adam? Is it left out as it is best to do for weight decay or is it integrated in the
         adaptive estimations?  I believe a plain SGD would be better justified when the
         goal is to understand an improvement to the flow of information in the network

We used Keras built-in support for additional losses through Layer.add\_loss. According to its source code, it is added to the  main loss, so it is integrated into the adaptive estimations. We chose Adam because its ubiquitous use in deep learning community.

Nevertheless, the choice of optimizer should have marginal effects in the experimentation, as the problems under study are independent from it.

2.3 ABOUT OUR EXPERIMENT PROTOCOL

Reviewer #3 states that:

        Results may be misleading both because learning rates are not adjusted, which
        can cause deeper models to diverge while they could still be trained and because
        number of epochs is limited instead of letting the models train until convergence. I
        note that plots comparing train and validation accuracy in Figure 1 shows better
        results on the validation set than on the training set for ReLU and ReLU + BN.
       This suggests indeed that the optimization diverged, breaking the training accu-
       racy while leaving the validation accuracy close to random.

We provide with activation plots in the Appendix Section A.3 (Figure 9) which prove that the activations are zero even at initialization and disproving the possibility of divergence. Also, the networks in the toy dataset are trained for $5000$ epochs, more than enough to achieve convergence.

Additionally, if the networks had diverged, we would see the networks failing at the same depth, whereas we observe failure at different depths (depending on the width) as expected. For instance, in the case of ReLU which (most prone to killing the dataset), readers can verify in Figures 1a and 1d that the networks start failing at 1x1 (depth x width), then at 10x2, then at 20x5 and finally at 30x20, following a sort of increasing pattern on the width. Also, in Figure 2a and 2d we see the same pattern: 20x4 dies, but 20x8 does not. For ReLU-BN (Figures 2b and 2e) the pattern is even more certain. These results support our conjecture that the networks fail because the dataset dies, by virtue of the large number of dead units and points, as predicted (see our Theorem 1 on dead points in the Appendix or Theorem 1 on dead units in Lu et al. (2019)), instead of showing divergence due to an unsuitable learning rate that would kill all the networks disregarding the width.

---

### Author Response · Authors · 2019-11-13
**1.4  ON THE RELATION BETWEEN CONSTRAINTS AND PARAMETER UPDATE**

1.4  ON THE RELATION BETWEEN CONSTRAINTS AND PARAMETER UPDATE

Both reviewers #1 and #3 request clarification of the transit between Equations (12) and (13). Notice that up to equation (12), we have only stated constraints upon the slack variables $\xi_{\pm}$ (that involve the dataset and parameters $\vec{w}$ and $b$). In Equation (13) we introduce the objective function to optimize the values of $\xi_{\pm}$. Namely, the sum of all $\xi$.

In between Equations (11) and (12) we refer the reader to Vapnik \& Cortes' Support Vector Networks work, where the process of scalarization is introduced to transform a constraint formulation into an optimization problem.

Since we have different indications of the $\xi$, we must define different objective functions for unit-wise constraints or $g_U$ of equation (13) and point-wise constraints or $g_P$ from Equation (15). We presented the construct for unit-based constraints (transit between Equations (12) and (13)) and extended the notation for point-wise constraints on equation (15).

If we wish to differentiate with regards to $\vec{w}$ and $b$, we must notice that $\nabla_{w}\xi_{\pm}^u$ as stated on must be defined by parts:
$$
\begin{equation}
    \nabla_{w}\xi_{-}^u =
    \begin{cases}
     \vec{x}_{-}, & p(\vec{x}_{-}) > -1 \\
     0, & \text{otherwise}\\
    \end{cases}
\end{equation}
$$
$$
    \frac{\partial\xi_{+}^u}{\partial{b}} =
    \begin{cases}
     1, & p(\vec{x}_{-}) > -1 \\
     0, & \text{otherwise}\\
    \end{cases}
$$
and for $\xi_{+}^u$:
$$
    \nabla_{w}\xi_{+}^u =
    \begin{cases}
     -\vec{x}_{+}, & p(\vec{x}_{+}) < 1 \\
     0, & \text{otherwise}\\
    \end{cases}
$$
$$
    \frac{\partial\xi_{+}^u}{\partial{b}} =
    \begin{cases}
     -1, & p(\vec{x}_{+}) < 1 \\
     0, & \text{otherwise}\\
    \end{cases}
$$
while under the unit-based formulation $\vec{x}_{\pm}$ is fixed by the geometry of $X$, for the point-wise formulation, we must have
$$
    \nabla_{w_{-}}\xi_{-}^p =
    \begin{cases}
     \vec{x}, & p(\vec{x}) > -1 \\
     0, & \text{otherwise}\\
    \end{cases}
$$
$$
    \frac{\partial\xi_{-}^p}{\partial{b}} =
    \begin{cases}
     1, & p(\vec{x}) > -1 \\
     0, & \text{otherwise}\\
    \end{cases}
$$
and for $\xi_{+}^p$:
$$
    \nabla_{w_{+}}\xi_{+}^p =
    \begin{cases}
     -\vec{x}, & p(\vec{x}) < 1 \\
     0, & \text{otherwise}\\
    \end{cases}
$$
$$
    \frac{\partial\xi_{+}^p}{\partial{b}} =
    \begin{cases}
     -1, & p(\vec{x}) < 1 \\
     0, & \text{otherwise}\\
    \end{cases}
$$
that is parallel to the loss gradient $\nabla_{w}L$.

---

### Author Response · Authors · 2019-11-13
**1.3  ON THE OPTIMIZATION  OF THE CONSTRAINTS**

Reviewer # 3 states that:

          It is unclear to me how the slack variables xi depends on the weights, and therefore how they can be minimized by
          adjusting the weights using a penalty. I believe this should be clarified between equations (12) and (13).

Notice that under our formulation, we are examining the set diameter of the preactivation of a dataset $X$. That is, given a unit $u$ with vector parameters $\vec{w}\in\mathbb{R}^n$ and $b\in\mathbb{R}$, its preactivation is given by the function
$$
 \vec{z} = p(\vec{x}) = \vec{w}\cdot\vec{x}+b
$$
given a finite discrete set $X\subset\mathbb{R}^n$ ( a compact set of $\mathbb{R}^n$), we can define the extreme values of $X$ with regards to $u$ as
$$
    z_{min}^u = \min\limits_{\vec{x}\in X}p(\vec{x})
$$
$$
    z_{max}^u = \max\limits_{\vec{x}\in X}p(\vec{x})\\
$$
on a similar argument, given a layer $\ell$ with units $u_1,u_2,\ldots,u_{n_k}$ with pre-activations denoted by $p_1,p_2,\ldots,p_{n_k}$, we can define the extreme values of layer $\ell$ provided with a fixed $\vec{x}\in X$ as
$$
    z_{min}^{p} = \min\limits_{j=1\ldots,n_k}p_j(\vec{x})
$$
$$
    z_{max}^{p} = \max\limits_{j=1\ldots,n_k}p_j(\vec{x})\\

$$
notice that since $X$ is a finite discrete set of $\mathbb{R}^{n_k}$,  $z_{min}^{\star},z_{max}^{\star}$ exist (where $\star$ stands for $u$ or $p$). This fact is instrumental in our Equations (10) to (12). Since $X$ is bounded, and we are working in the real number system, we can find numbers $\xi_{+}^{\star}, \xi_{-}^{\star}$ such that
$$
    z_{min}^{\star} \leq -1+\xi_{-}^{\star}
$$
$$
    z_{max}^{\star} \geq  1-\xi_{+}^{\star}
$$
notice that the parameters $\vec{w}\in\mathbb{R}^{n_k}$ and $b$ are free in the definition of $z^{u}$, while $\vec{x}$ is free in the definition of $z^{p}$. In this sense, for $\xi_{\pm}^p$ we can find $\vec{w}_{\pm}^{p}\in\mathbb{R}^{n_k}$ and $b_{\pm}^{p}\in\mathbb{R}$ such that
$$
    \xi_{+}^{p} = \max(1 - (\vec{w_{+}}\cdot\vec{x} + b_{+}), 0)
$$
$$
    \xi_{-}^{p} = \max(1 + (\vec{w_{-}}\cdot\vec{x} + b_{-}), 0)\\
$$
(We number this equation as Equation 8 for future reference)

that constraints all possible configurations of parameters in units of $\ell$.
Naturally, $\xi_{\pm}$ depend on $\vec{w}$ and $b$. This is at the heart of the constraint formulation: choosing $\vec{w}$ and $b$ so that $z_{min}<0$ and $z_{max}>0$.

Meanwhile, given a fixed selection of $\vec{w}$ and $b$, since $X\subset\mathbb{R}^{n_k}$ is discrete and bounded, we must have that there exist $\vec{x}_{+}$ and $\vec{x}_{-}$ points of $X$ such that
$$
    \xi_{-}^{u} \leq \max(1 - (\vec{w}\cdot\vec{x}_{-} + b), 0)
$$
$$
    \xi_{+}^{u} \leq \max(1 + (\vec{w}\cdot\vec{x}_{+} + b, 0)
$$
(We number this equation as Equation 9 for future reference)

We wish to stress our geometric intuition, beyond calculations. We seek a combination of $\vec{w}$ and $b$ so that they cut through $X$ in the case of Sep-U, noticing that the preactivation $p$ defines a signed distance from the separating hyperplane $p(\vec{x})=0$.

The same argument is used in the field of Support Vector Machines, as cited in our references, particularly  Vapnik \& Cortes' Support Vector Networks from 1995 and  Boyd \& Vandenberghes' Convex Analysis from 2004.

---

### Author Response · Authors · 2019-11-13
**1.2 ON THE FORMULATION OF THE CONSTRAINTS**

Since our rationale attacks the dead unit/dead point problem using
geometric arguments, we found beffiting  to  use  a  classical  mathematical  notation.To  be  precise,  we  used  the  notation  of  J.R.Munkres’ Analysis on Manifolds.

We are surprised to see the comments of two reviewers concerned with the vector $\hat{\mathbf{e}}$, that is the $one-hot$ canonical basis vector of $\mathbb{R}^n$, the cornerstone of linear algebra.

As stated at the start of section 2 (Characterizing dead units and dead points), we regard layer functions $\ell$ as vector-valued functions. Thus, instead of the cumbersome tuple notation for vectors:
$$
(u_1,u_2,\ldots,u_{n_k})\in\mathbb{R}^{n_k}
$$
we used a compact notation based on the observation that every vector in $\mathbb{R}^n$ can be written as a linear combination of the canonical vectors
$$
    \hat{\mathbf{e}}_j = (0,\ldots,0,1,\ldots,0)
$$
where the $1$ is located on the $j$-th position. As a consequence, we can introduce superscripts and subscripts (to index layers and units within them) that allows us to describe dead units and dead points using queries on the indices, as remarks 2.1 (Dead unit), 2.2 (Dead point within a layer) and 2.3 (Dead point for the network) attest. Such approach is then applied in our formulation in Equations (8) to (11).

We observe from the minor comments of Reviewer #3 that there is also a misinterpretation of equation (8), as reviewer #3 claims that equation (8) has an additional (misplaced) term:
$$
    R_X(u_j^k) \equiv \emptyset \neq \{upper(u_j^k)\cap X\}\neq X
$$
as stated in the preceding paragraph, Equation 8 is a predicate and not an assignment. That is $R_X(u_j^k)$ has an assigned truth value (false or true), depending on whether $upper(u_j^k)$ is a non-empty proper subset of $X$.

In our construction, separating units cannot be dead (in the sense of Remark 2.1), but neither can it map non-trivially the whole dataset $X$. As a consequence, it is of extreme importance that $upper(u_j^k)$ remains a proper subset of $X$, see Figures 1, 2 and 3 (We apologize for the poor quality of the figures, we provide with the latex version with the code).

                                     ^
                                     |
                                     |                                           ^
                                     |                                           |
                +---------------------------------+                   |                            +----------------+
                       +--------------------+              +---------------------+              |                    |
                       |                          |              |             |            |               |         ^         |
                       |      DATASET    |      +----------------+-------------- +        |         |         |
                       |                          |              |                           |               +---------------+
                       |                          |              |       DATASET    |                          |
                       +--------------------+              +---------------------+        +------------+------------+

                    Figure 1: Dead unit         Figure 2: Separating unit      Figure 3: Affine unit

We apologize if the $\equiv$ sign is misleading, it is a substitute for if and only if or notation-wise $\Leftrightarrow$ and it was borrowed from Gries \& Scheneiders'  Logical Approach to Discrete Math from 1996.

With regards to remark 2.1 (definition of a dead unit) we apologize for the lack of clarity, it does require us to introduce the notion of hidden representation of a dataset, and its image at the $k$-th layer.

---

### Author Response · Authors · 2019-11-13
**1.1 ON THE GOAL AND METHOD  OF THE ARTICLE**

1.1 ON THE GOAL AND METHOD  OF THE ARTICLE

We are surprised to see how none of the reviewers made any comments about the goal of the paper (presented on the second and third paragraphs of the introduction):  removing the need for model width scaling, particularly, when it is written in the title itself.

While  the  goal  of  the  article  is  to  train  deeper  networks  without  resorting  to  addition  of units/neurons (the model width scaling technique,  see Huang et al. (2016),  or Tan & Le (2019), our constraint formulation is meant to solve what we determine that is the root of the problem, the presence of dead points/units (while extending the notion of dead units to dead points).

While  we  appreciate  the  feedback  on  our  experiment  protocol,  we  find  it  curious  that  the  critique on our work is centered around experimentation, but not on the soundness of our rationale.

Since  this  paper  is  written  following  a  mathematical  reasoning,  experiments  serve  as illustration rather than justification, and our experimentation is consistent with our formulation.

We build on Lu et al. (2019), conjecturing that model scaling is used to circumvent the problem of incorrect hyperplane placement (of ReLU units) during initialization, by adding units that hopefully will accomodate for the dataset favorably (see the third paragraph of our introduction).   This is instrumental, and is made evident in our remarks 2.1 and 2.2 (dead units and dead points), under our definition the $lower$ and $upper$ sets of Equation 3.

With  these  remarks  in  mind,  it  felt  natural  to  propose  constraints  to  alleviate  the  problem,
based on the fact that data-sets are finite (but rather large) discrete sets of $\mathbb{R}^n$

---

### Author Response · Authors · 2019-11-13
**Superseding Model Scaling by Penalizing Dead Units and Points with Separation Constraints: A Rebuttal on Feedback received for ICLR 2020**


*******IMPORTANT*******
We have added the PDF version of this rebuttal into the zip of the code for your reading pleasure.
************************

SUPERSEDING MODEL SCALING BY PENALIZING DEAD UNITS  AND POINTS  WITH SEPARATION CONSTRAINTS: A REBUTTAL ON FEEDBACK RECEIVED FOR ICLR 2020

TABLE OF CONTENTS
1    On the Theoretical aspects of our articles
    1.1    On the Goal and Method of the article
    1.2    On the formulation of the constraints
    1.3    On the Optimization of the Constraints
    1.4    On the relation between constraints and parameter update

2    About Technical aspects of the Constraint Formulation and our Experiments
    2.1    Choice of the dataset
    2.2    On our Choice of Optimizer
    2.3    About our Experiment Protocol
    2.4    About our Colormaps
    2.5    About our approach on Zero initialization
    2.6    On our too Simple Tasks
    2.7    Empirical evidence of Dead Points
    2.8    Use of convolutional layers
    2.9    On the Convergence of the proposed method
    2.10  About Testing the Separation Constraints on VGG or ResNet

3    Specific comments that we wish to address

References


1. ON THE THEORETICAL ASPECTS OF OUR ARTICLES

In this section we wish to address several aspects of our article that cover comments from several reviewers concerning the theoretical component of our article over which we wish to clarify.  We will first address the comments on the reviewers about the purpose, method and notation used in our article (subsection 1.1), followed by our response to the shared concerns of the reviewers on the constraint formulation (subsection 1.2) and finish on our comments on how the training of a network is affected by the introduction of constraints (subsection 1.4).

---

### Author Response · Authors · 2019-11-13
**TL;DR**

Following an email received from ICLR 2020 asking for "concise responses" and realizing the current extension of the rebuttal well past 8 pages, we have decided to post a tl;dr summarizing the main points in order to hopefully receive an answer (and discuss it) interactively before the rebuttal period ends.



We are surprised to see how none of the reviewers made any comments about the goal of the paper (presented on the second and third paragraphs of the introduction):  removing the need for model width scaling, particularly, when it is written in the title itself. Instead, they focused solely on the dead unit/point problem, which is instrumental. This made the reviewers biased against our interest in the problems derived from dead points/units and depth. Consequently, the reviewers complained about the choice of dataset being too simple, without realizing that we needed a dataset as simple as possible in order to isolate error from other sources as underfitting or overfitting nor realizing that the the dependence of width in depth is a intrinsic property of neural networks: it is invariant to dataset (Lu et al., 2019, Theorem 1).

We disprove Reviewer #3' complaint about the learning rate causing divergence due depth, pointing out that if that were the case, the networks would fail at the same number of layers. However, death in networks is proportional to their width as we predicted. We also provide with activation plots where the points cluster at zero or in few points, sustaining our claim. We address the criticism of not comparing our proposal to ReLU and ReLU-BN or Annealed Dropout when testing Zero Initialization by evidencing that they cannot function with Zero Initialization anyway. We find Reviewer #2's requests to be simply out of the scope of this already bloated article, but we compromise: we have added an explanation of the transient states observed during training (in the Appendix). We clarify several misunderstandings and doubts, by elaborating on the internal representations, and on how to derive the gradient of the Separation Constraints. We also provide a theorem proving the existence of dead points asymptotically (also in the Appendix).

---

> ### Comment · AnonReviewer3 · 2019-11-19
> **Answer to rebuttal #1**
>
> I'd like to first recognize the great efforts of the authors to give a very detailed rebuttal. I do appreciate all the corrections and clarifications.
>
> > 1.1
> > Since  this  paper  is  written  following  a  mathematical  reasoning,  experiments  serve  as illustration rather than justification, and our experimentation is consistent with our formulation
>
> I absolutely disagree. The mathematical formulation is built on assumptions that can hardly be verified in other ways than empirical investigations. However sound the mathematical reasoning is, if the assumptions are wrong the experiments will demonstrate it.
>
> > We are surprised to see the comments of two reviewers concerned with the vector , that is the  canonical basis vector of , the cornerstone of linear algebra.
>
> I still fail to appreciate the usefulness of the notation with the basis vector in this context versus simply representing u_j^k.
>
> > We observe from the minor comments of Reviewer #3 that there is also a misinterpretation of equation (8), as reviewer #3 claims that equation (8) has an additional (misplaced) term:
>
> I would remark that there is indeed a mistake, as the paper's equation has a subset symbol instead of inequality symbol. Given that this symbol can both represent subset or proper subset, without any clarification this seems like a mistake. I understand now the authors implied proper subset, anyhow the inequality symbol is more explicit.
>
> > 1.3 - 1.4
>
> I strongly appreciate the detailed explanation and noted these have been added to the appendix as well.
>
> > 2.1
>
> I disagree with the authors that more complex datasets would defeat the purpose of experimentation and will explain below.
>
> > Since we are interested in the problems arising from dead points and units when the networks grow deeper, we need a dataset a simple as possible in order to isolate error from other sources as underfitting or overfitting.
>
> This is turning an instrumental goal into a final one. The objective of the paper is to allow training of deep networks without the need for width scaling. Implicitely, we assume that this should be done without harming generalization performance of models, otherwise it would be better advised to keep using width scaling. Therefore, control over the dead points and units is the instrumental goal for the ultimate goal which is to allow training of deep networks more efficiently (without width scaling). If we focus on the dead points and units only while factoring out overfitting/underfitting, we are turning it into the ultimate goal.  The experimentations of the paper *must* be tied to the concepts of overfitting/underfitting. Also, given the formulation proposed to alleviate dead points and units, an additional loss function, ignoring overfitting/underfitting would be ignoring an important effect of the intervention.
>
> > We are only concerned with alleviating dead points and units to enable back-propagation, nothing more. That is why we chose a very simple dataset which can be solved with three units, to show that if the network fails, it must be due to depth-induced problems.
>
> That I agree with, but there is an important piece that is missing. If we want to reduce width scaling, it is because deep architectures (which required width scaling) has proved to be performing better than shallow ones on some tasks otherwise we would still be using shallow models. If there is an interest in reducing the computational cost of width scaling, it is specifically on these tasks. Proving that the proposed method does allow training of deep models on a simple toy task is, I agree, an important first step in the experimentation. Nevertheless, it is insufficient to confirm that it is not harming the ultimate goal. This is also where I strongly disagree with the authors on the role of experimentation as a mere illustration. The nature of these tasks of interest where deep architectures proved to perform better would be quite a challenge to encapsulate into assumptions that are well defined mathematically. We need the experiments to validate it.
>
> > Additonally, the complexity on the number of constraints is (theoretically) low, but we suspect that our implementation does not reuse the activations [...]
>
> This is why I believe synthetic data would be a more promising solution than brute-force benchmarking on large datasets such as ImageNet. The creation of the synthetic data would not be triavial however.
>
> > 2.2
>
> Given that performance gains can be obtained by factoring weight decay out of the adaptive part of Adam, I suspect that the contributed loss may also behave differently if integrated in the adaptive part or not. Therefore, I am not confident that the choice of optimizer have marginal effect. Using stochastic gradient descent would leave no doubt. I acknowledge however that this is a minor issue.

---

> ### Comment · AnonReviewer3 · 2019-11-19
> **Answer to rebuttal #2**
>
>
> (comments are in reverse order)
>
> > 2.3
>
> 5000 epochs should be enough indeed.
>
> In short, what I meant is that controlling the learning rate by fixing it instead of optimizing it will likely lead to an exacerbated difference between ReLU and other results. I don't believe it would turn the results upside-down, but the difference is exacerbated.
>
> > 2.4
>
> The values were mentioned in the captions indeed. My mistake.
>
> > 2.5
>
> Agreed.
>
> > 2.6
>
> Overfitting is function of both the generalization error and the capacity of the model. If increasing the capacity leads to increased generalization error while training error decreases, then the model is overfitting. Training accuracy is higher at depth 10 than 2 for Sep-UP, while the validation accuracy is lower at depth 10 than depth 2. For all the other cases, both training accuracy and validation accuracy are dropping. This is again my main concern with the experiments. There is no point in increasing depth if it harms generalization. The innocuity of the loss function must be confirmed on tasks where there is something to gain by increasing depth. This is inline with my comments to point 2.1 above.
>
> The results in figure 1 and 2 can be said to qualitatively confirm that Sep-UP does lead to comparable results. To which degree in particular is difficult to assert given the large scale of the colors. Anyhow, there is no measurable gain obtained by increasing the depth. Again, there is no point in increasing the depth.
>
> > 2.7-2.8
>
> Thank you for the clarifications.
>
> > 2.10
>
> I agree there is no need for very large models or dataset such as ImageNet, as long as the task at hand is better solved by deep models.
> ----
>
> To conclude, I still believe this paper lacks empirical evidence that the proposed loss would help train deep models without harming performance on tasks where depth matters. In light of all the corrections and clarifications, I upgrade my vote to weak reject.

---

### Decision · Program_Chairs · 2019-12-19

**Decision:**

Reject

**Comment:**

This paper proposes constraints to tackle the problems of dead neurons and dead points. The reviewers point out that the experiments are only done on small datasets and it is not clear if the experiments will scale further. I encourage the authors to carry out further experiments and submit to another venue.

---

> ### Comment · AnonReviewer3 · 2019-12-20
> **Clarification**
>
>
> Although I agree with the decision, I believe the justification is wrong.
>
> I am fairly confident that the observations would scale to larger datasets, that is, deep models without width correction are made trainable with the proposed constraint. The issue in the paper is the experimental setting, in which depth gives no advantages over large shallow networks. For instance, reported performances on CIFAR10 are significantly lower than current best performing models (VGGs, ResNets) and are on par with shallow MLPs. I do agree with authors that they should not work on such architectures directly, since these have been architected with increasing width, but the proposed constraint should make it possible to reach closer performances with deep networks. Since deeper models are currently the best performing models on CIFAR10, the efficiency of a constraint that enables training of deep models without harming their performance could be demonstrated on this dataset. CIFAR10 is enough, larger datasets would be a waste of computations.